

# Nine years of global hydrocarbon emissions based on source inversion of OMI formaldehyde observations

Maite Bauwens[1], Trissevgeni Stavrakou[1], Jean-François Müller[1], Isabelle De Smedt[1], Michel Van Roozendael[1], Guido R. van der Werf[2], Christine Wiedinmyer[3], Johannes W. Kaiser[4], Katerina Sindelarova[5,6], and Alex Guenther[7]

[1]Royal Belgian Institute for Space Aeronomy, Avenue Circulaire 3, 1180, Brussels, Belgium

[2]Vrije Universiteit Amsterdam, Faculty of Earth and Life Sciences, Amsterdam, The Netherlands

[3]National Centre for Atmospheric Research, Boulder, CO, USA

[4]Max Planck Institute for Chemistry, Mainz, Germany

[5]UPMC Univ. Paris 06; Université Versailles St-Quentin; CNRS/INSU; LATMOS-IPSL, Paris, France

[6]Charles University in Prague, Department of Atmospheric Physics, Prague, Czech Republic

[7]University of California, Irvine, USA

*Correspondence to:* Maite Bauwens (maite.bauwens@aeronomie.be)

**Abstract.**

Formaldehyde (HCHO) being a high-yield product in the oxidation of most volatile organic compounds (VOCs) emitted by fires, vegetation and anthropogenic activities, satellite observations of HCHO are well-suited to inform us on the spatial and temporal variability of the underlying VOC sources. The long-record of space-based HCHO column observations from

5    the Ozone Monitoring Instrument (OMI) is used to infer emission flux estimates from pyrogenic and biogenic volatile organic compounds (VOCs) on the global scale over 2005-2013. This is realized through the method of source inverse modelling, which consists in the optimization of emissions in a chemistry-transport model (CTM) in order to minimize the discrepancy between the observed and modelled HCHO columns. The top-down fluxes are derived in the global CTM IMAGESv2 by an iterative minimization algorithm based on the full adjoint of IMAGESv2, starting from a priori emission estimates provided by the newly

10   released GFED4s (Global Fire Emission Database, version 4s) inventory for fires, and by the MEGAN-MOHYCAN inventory for isoprene emissions. The top-down fluxes are compared to two independent inventories for fire (GFAS and FINNv1.5) and isoprene emissions (MEGAN-MACC and GUESS-ES).

   The inversion indicates a moderate decrease (ca. 20%) of the average annual global fire and isoprene emissions, from 2028 TgC in the a priori to 1653 TgC for burnt biomass, and from 343 to 272 Tg for isoprene fluxes. Those estimates are

15   acknowledged to depend on the accuracy of formaldehyde data, as well as on the assumed fire emission factors and the





oxidation mechanisms leading to HCHO production. Strongly decreased top-down fire fluxes (30-50%) are inferred in the peak fire season in Africa, and during years with strong a priori fluxes associated to forest fires in Amazonia (in 2005, 2007 and 2010), bushfires in Australia (in 2006 and 2011), and peat burning in Indonesia (in 2006 and 2009), whereas generally increased fluxes are suggested in Indochina and during the 2007 fires in Southern Europe. Moreover, changes in fire seasonal patterns are suggested, e.g. the seasonal amplitude is reduced over Southeast Asia. In Africa, the inversion indicates increased fluxes due to agricultural fires, and decreased maxima when natural fires are dominant. The top-down fire emissions are much better correlated with MODIS fire counts than the a priori inventory in regions with small and agricultural fires, indicating that the OMI-based inversion is well-suited to assess the associated emissions.

Regarding biogenic sources, significant reductions of isoprene fluxes are inferred in tropical ecosystems (30-40%), suggesting overestimated basal emission rates in those areas in the bottom-up inventory, whereas strongly positive isoprene emission updates are derived over semi-arid and desert areas, especially in Southern Africa and Australia. This finding suggests that the parameterization of the soil moisture stress used in MEGAN greatly exaggerates the flux reduction due to drought in those regions. The isoprene emission trends over 2005-2013 are often enhanced after optimization, with positive top-down trends in Siberia (4.2%/yr) and Eastern Europe (3.9%/yr), likely reflecting forest expansion and warming temperatures, and negative trends in Amazonia (-2.1%/yr), South China (-1%/yr), the United States (-3.7%/yr), and Western Europe (-3.3%/yr), which are generally corroborated by independent studies, yet their interpretation warrants further investigation.

# 1 Introduction

Complementary to bottom-up methodologies for deriving emissions estimates, inverse modelling has the potential to improve those estimates through the use of atmospheric observations of trace gas compounds, in particular over regions undergoing fast economic development and facing intense air pollution problems, like Eastern China (Worden et al., 2012; Reuter et al., 2014; Mijling and van der A, 2012; Ding et al., 2015), but also on the global scale (Jaeglé et al., 2005; Chang and Song, 2010; Kopacz et al., 2010). Pollutants like CO and $NO_2$ are directly detected from satellite and their emissions have been inferred using inversion techniques at different scales (e.g. Pétron et al. (2004); Müller and Stavrakou (2005); Stavrakou et al. (2008); Kopacz et al. (2010); Tang et al. (2013)). The detection of formaldehyde columns from satellite sensors measuring in the UV-Visible spectral window opened the way for the derivation of fluxes of non-methane volatile organic compounds (NMVOCs), a broad class of formaldehyde precursors emitted by vegetation, fires and anthropogenic activities (Chance et al., 2000; Palmer et al., 2003, 2006). These compounds have a profound impact on air quality and climate, owing to their influence on OH levels and the methane lifetime and to their role as precursors of ozone and secondary organic aerosols (Hartmann et al., 2013). The accurate estimation of their fluxes is therefore of utmost importance.

Natural emission from vegetation is the dominant VOC source. The global annual flux is estimated at ca. 1000 Tg VOC, with isoprene accounting for half of this emission (Guenther et al., 1995, 2012; Sindelarova et al., 2014). Despite a general consensus on the isoprene emission patterns, including their dependence on temperature and light density responsible for their marked diurnal and seasonal variations, these emission estimates come, however, with large uncertainties, associated



with the strong variability of emission factors and the extrapolation of sparse measurements to larger scales. An uncertainty of a factor of 2 in global and regional isoprene fluxes was reported based on a compilation of numerous literature studies (Sindelarova et al., 2014), whereas emission models were found to be strongly sensitive to choices of input variables, leading to even wider uncertainty, ca. 200-1000 TgC/yr globally (Arneth et al., 2011).

The global biomass burning fluxes are estimated by bottom-up inventories at ca. 1300-2200 TgC on a yearly basis, which corresponds to 40-100 TgVOC/yr using emission factors from the compilation of Andreae and Merlet (2001) or Akagi et al. (2011) (van der Werf et al., 2010; Wiedinmyer et al., 2011). These estimates, however, depend on assumptions made in fire emission models regarding fuel loading and consumption efficiency, and on the quality of land cover maps and fire proxies from satellite (Hyer and Reid, 2009; Wiedinmyer et al., 2011; Soares et al., 2015).

Formaldehyde is a high-yield product in the oxidation of a large majority of NMVOCs. Isoprene alone is responsible for approximately 30% of the global HCHO burden according to model estimates (Stavrakou et al., 2009b), whereas the contribution of vegetation fires is globally small (3%), but can be locally very important. Spaceborne vertical columns of HCHO retrieved from GOME, SCIAMACHY, OMI and GOME-2 sensors have been used to constrain the VOC budget at different scales (e.g. Palmer et al. (2003, 2006); Millet et al. (2008); Barkley et al. (2013); Bauwens et al. (2014); Zhu et al. (2014)).

Top-down flux estimates deduced from two satellite sensors with different overpass times showed a good degree of consistency over the Amazon (Barkley et al., 2013) and globally (Stavrakou et al., 2015). The latter study using GOME-2 (9:30 LT) and OMI (13:30 LT) HCHO observations in 2010 reported a good agreement between the inversion results over most areas and identified large regions where the derived emissions were highly consistent (e.g. Amazonia, Southeastern US). Encouraged by those results, and relying on a multi-year record of HCHO columns observed by the OMI sensor, we use inverse modelling to

derive top-down pyrogenic and biogenic VOC estimates over 2005-2013. The satellite data offer an unparalleled opportunity to bring new insights in our understanding of emissions and their quantification, to infer long-term seasonal and interannual flux variability, and to detect potential emission trends that might not be well represented in bottom-up inventories. To this purpose, we use a global CTM, coupled with an inversion module and a minimization algorithm adjusting the emissions used in the model in order to achieve an optimal match between the modelled and the observed HCHO columns while accounting for

errors on the a priori emissions and the HCHO observations. The optimized fluxes are compared with independent bottom-up pyrogenic and biogenic emission inventories as well as with previous literature studies. The methodology is briefly presented in Sect. 2, and an overview of the results is discussed in Sect. 3. The top-down fluxes and comparisons to bottom-up inventories over big world regions are discussed thoroughly in Sect. 4- 8 and emission trends in Sect. 9. Conclusions and final remarks are presented in Sect. 10.

## 2   Methods

We used formaldehyde observations retrieved from the OMI spectrometer aboard the Aura mission and fully documented in a recent study (De Smedt et al., 2015). The retrievals are based on an improved DOAS algorithm that reduces the effect of interferences between species and ensures maximum consistency between the OMI and GOME-2 columns. The current data



version (v14) uses an iterative algorithm to remove spikes in the residuals of the slant columns and a procedure based on the background normalisation to remove striping artefacts due to calibration problems (Boersma et al., 2011; Richter et al., 2011; De Smedt et al., 2015). In addition to the destriping procedure, in order to reduce the effect of the OMI row anomaly issue affecting the spectra after 2007 (http://www.knmi.nl/omi/research/product/rowanomaly-background.php), the OMI rows present-

ing higher levels of noise and fitting residuals than the average were systematically removed from the dataset (De Smedt et al., 2015). Although this filtering leads to a loss of coverage, the resulting dataset is more appropriate for addressing trend studies, as explained in De Smedt et al. (2015).

The IMAGESv2 global model calculates the concentrations of 131 transported and 41 short-lived trace gases with a time step of 6 hours at $2° \times 2.5°$ resolution between the surface and the lower stratosphere. The effect of diurnal variations is

accounted for through correction factors on the photolysis and kinetic rates obtained from model simulations with a time step of 20 minutes, which are also used to calculate the diurnal shapes of formaldehyde columns required for the comparison with satellite data. A detailed model description is provided in Stavrakou et al. (2013). Meteorological fields are obtained from ERA-Interim analyses of the European Centre for Medium-range Weather Forecasts (ECMWF). The model uses anthropogenic NOx, CO, $SO_2$, $NH_3$ and total NMVOC emissions from the Emission Database for Global Atmospheric Research (EDGAR4.2,

http://edgar.jrc.ec.europa.eu), which is overwritten by the EMEP inventory (http://www.ceip.at/ms) over Europe, and by the REASv2 inventory (Kurokawa et al., 2013) over Asia. The NMVOC speciation is obtained from REASv2 over Asia and from the RETRO inventory (Schultz et al., 2007) elsewhere. The emissions over the US are scaled according to the NEI national totals for all years between 2005 and 2013 (http://www.epa.gov/air-emissions-inventories/air-pollutant-emissions-trends-data).

Biomass burning emissions are taken from the latest version of the Global Fire Emissions Database, GFED4s (July 2015),

which includes the contribution of small fires based on active fire detections (Randerson et al., 2012; Giglio et al., 2013). The GFED data are available on a daily basis at $0.25° \times 0.25°$ resolution from 1997 through the present at http://www.globalfiredata.org. Those emissions are distributed vertically according to Sofiev et al. (2013).

A priori isoprene emissions are obtained from the MEGAN-MOHYCAN model (Müller et al., 2008; Stavrakou et al., 2014) for all years of the study period at a resolution of $0.5° \times 0.5°$ (http://tropo.aeronomie.be/models/isoprene.htm). Besides the

emission dependence on leaf temperature, photosynthetically active radiation (PAR), leaf area and leaf age, the model accounts for the inhibition of isoprene emissions in very dry soil conditions through a dimensionless soil moisture activity factor ($\gamma_{SM}$) expressed as a function of volumetric soil moisture content (Guenther et al., 2006) obtained from the ERA-Interim reanalysis. The parameterization of $\gamma_{SM}$ bears large uncertainties, as it is based on scarce (and sometimes contradictory) field data, and its implementation can lead to very different results depending on the choice of database for soil moisture data (Müller et al., 2008;

Sindelarova et al., 2014; Marais et al., 2012). It reduces the emissions by ca. 20% globally according to MEGAN-MOHYCAN, with strongest effects (up to factor of 3 or more) over Australia and Southern Africa, and to a lesser extent over Northern Africa (Sahel), the Western US and the Middle East.

The chemical degradation mechanism of pyrogenic NMVOCs is largely described in Stavrakou et al. (2009a), with only minor modifications. This mechanism includes an explicit treatment for 16 pyrogenic formaldehyde precursors. The emissions

of other pyrogenic compounds is represented through a lumped compound (OTHC) with a simplified oxidation mechanism





designed in order to reproduce the overall formaldehyde yield of the explicit NMVOC mix it represents. The oxidation mechanism for isoprene is based on Stavrakou et al. (2010), modified to account for the revised kinetics of isoprene peroxy radicals according to the Leuven Isoprene Mechanism version 1 (LIM1) (Peeters et al., 2014), as well as for the chemistry of the isoprene epoxides (IEPOX) following the Master Chemical Mechanism MCMv3.2 (http://mcm.leeds.ac.uk/MCMv3.2/). The

formaldehyde yield in isoprene oxidation by OH is calculated using a box model to be 2.4 mol/mol in high NOx (1 ppbv $NO_2$, after 2 months of simulation) and 1.9 mol/mol for 0.1 ppbv $NO_2$. It should be stressed that the isoprene mechanism still bears important uncertainties at low NOx conditions, as both the oxidation products of the isoprene epoxides and the isomerisation products of isoprene peroxy radicals have complex degradation mechanisms that are still far from being well elucidated despite recent progress (Peeters et al., 2014; Bates et al., 2016). Note that suppressing the isomerization channel in the isoprene

degradation resulted in only slightly higher model HCHO columns over isoprene-rich regions (Stavrakou et al., 2015).

The mismatch between the CTM and the observations, quantified by the cost function $J$,

$$J(\mathbf{f}) = \frac{1}{2}\big((H(\mathbf{f}) - \mathbf{y})^T \mathbf{E}^{-1}(H(\mathbf{f}) - \mathbf{y}) + \mathbf{f}^T \mathbf{B}^{-1}\mathbf{f}\big) \qquad (1)$$

is minimized through an iterative quasi-Newton optimization algorithm, which is based on the calculation of the partial derivatives of $J$ with respect to the input variables, in our case are scalar variables $\mathbf{f} = (f_j)$, such that the optimized flux can be

expressed as

$$\Phi_i^{opt}(x,t) = \sum_{j=1}^{m} e^{f_j} \Phi_i(x,t), \qquad (2)$$

with $\Phi_i(x,t)$ being the initial flux depending on space (latitude, longitude) and time (month), and $m$ the emission categories/processes. In Eq. 1, $H(\mathbf{f})$ denotes the model acting on the variables, $\mathbf{y}$ the observation vector, $\mathbf{E}$ and $\mathbf{B}$ the covariance matrices of the errors on the observations and on the a priori parameters $\mathbf{f}$, respectively, and $^T$ means the transpose of the

matrix. The partial derivatives of $J$ with respect to $\mathbf{f}$ are calculated by the discrete adjoint of IMAGESv2 chemistry-transport model (CTM) (Müller and Stavrakou, 2005; Stavrakou et al., 2009b). The derivation of monthly pyrogenic and biogenic fluxes is carried out on global scale at the resolution of the model ($2° \times 2.5°$), as described in detail in Stavrakou et al. (2015). The inversions are performed separately for all years of the study period (2005-2013), and about 60,000 flux parameters are optimized per year globally.

The covariance matrix of the observational errors is assumed diagonal. The errors are calculated as the squared sum of the retrieval error and a representativity error set to $2\times10^{15}$ molec.cm$^{-2}$. The assumed error on the a priori biogenic and pyrogenic fluxes is factor of 3. The spatiotemporal correlations among the a priori errors on the flux parameters are defined as in Stavrakou et al. (2009b). About 20-40 iterations are needed to reach convergence, which is attained when the gradient of the cost function is reduced by a factor of 1000 with respect to its initial value. The cost function generally decreases by ca.

45-55% in comparison to its initial value.

Figure 1 illustrates a comparison between observed monthly mean HCHO column densities over 2005-2013 and monthly columns simulated by the IMAGESv2 model sampled at the time and location of the satellite measurement. The observed monthly averages exclude scenes with cloud fractions higher than 40% and land fractions lower than 20%, as well as data with





a retrieval error higher than 100%. The number of effective observational constraints is highest in the first years of the OMI mission (ca. 17,000 per year), and declines by about 15% after 2009 due to instrumental degradation effects (De Smedt et al., 2015), whereas the data availability is higher during the summer than in the winter in the Northern Hemisphere (ca. 1600 vs. 1200 measurements per month). The satellite columns are freely available at the BIRA-IASB website (http://h2co.aeronomie.be).

The OMI-based emission fluxes presented in this study are available at the GlobEmission web portal (http://www.globemission.eu).

## 3  Overview of the results

The source optimization leads to a good overall agreement with the OMI observations (Fig. 1), in particular in the Tropics, as a result of the high signal-to-noise ratio in the observations at these latitudes. The a posteriori columns remain close to the a priori at high latitudes, mainly due to lower data availability and higher observational errors at these latitudes (De Smedt et al., 2015).

The inferred mean HCHO columns over the study period are generally decreased by 20-25% over the Amazon and Equatorial Africa, whereas a mean decrease of about 13% is found in the Southeastern US during summertime (cf. Supplement, Fig. S1). The HCHO columns are increased in a few regions after inversion, especially during biomass burning events. The annually averaged global distribution of pyrogenic and isoprene emissions over 2005-2013 before and after optimization is illustrated in Fig. 2. Fig. 3 displays the extent of the regions over which comparisons will be discussed. Bottom-up and top-down emission

estimates are summarized in Table 1 and 2.

The OMI-based fire flux estimates are compared with two independent inventories GFAS and FINNv1.5. The Global Fire Assimilation System (GFAS) is based on assimilation of fire radiative power observed from the MODIS instruments aboard the Terra and Aqua satellites (Kaiser et al., 2012) and provides daily global fire emission estimates at $0.5° \times 0.5°$ and $0.1° \times 0.1°$ resolution for 2003 onwards (http://eccad.sedoo.fr). The Fire Inventory from NCAR (FINN) version 1.5 is an updated version

of the FINN daily global high-resolution inventory (Wiedinmyer et al., 2011) available at http://bai.acd.ucar.edu/Data/fire. In addition to GFED4s, we also used GFED4. Both version have adopted lower fuel consumption rates than the previous version GFED3 (van der Werf et al., 2010) to better match field observations (Leeuwen et al., 2014), but in GFED4s this decrease is compensated for by the addition of small (s) fire burnt area. It is available at http://www.globalfiredata.org. The average 2005-2013 global burnt biomass is estimated at 1938, 2006, and 1438 TgC/yr, in GFAS, FINNv1.5 and GFED4, respectively

(Table 1, Fig. S2).

The isoprene emission estimates are compared to two bottom-up inventories, MEGAN-MACC and GUESS-ES (Fig. S3). MEGAN-MACC (Sindelarova et al., 2014) relies on the MEGANv2.1 model for biogenic volatile organic compounds (BVOC) and is based on the MERRA reanalysis fields (Rienecker et al., 2011). The emissions are provided at $0.5° \times 0.5°$ resolution and on a monthly basis from 1980 through 2010. The GUESS-ES isoprene inventory is based on the physiological isoprene

emission algorithm described by Niinemets et al. (1999) and updated by Arneth et al. (2007). It is coupled to the dynamic global vegetation model LPJ-GUESS (Sitch et al., 2003) and is driven by the CRU (Climatic Research Unit) monthly meteorological fields (Mitchell and Jones, 2005) at $1° \times 1°$ resolution between 1969 and 2009. Both inventories are available from the ECCAD data portal (http://eccad.sedoo.fr). The mean isoprene emission amounts to 452 Tg/yr and to 570 Tg/yr, accord-



ing to the GUESS-ES and to MEGAN-MACC inventory, respectively (Table 1), and both lie much higher than the a priori MEGAN-MOHYCAN inventory (343 Tg/yr average over 2005-2013). The large discrepancy between MEGAN-MACC and MEGAN-MOHYCAN datasets, both relying on the MEGAN emission model (Guenther et al., 2006) and the same version of basal emission factors (version 2011) can be explained to a large extent by (i) the neglect of soil moisture stress effects in

MEGAN-MACC, (ii) a reduction by a factor of 4.1 of the basal emission factors for forests in Asia in MEGAN-MOHYCAN (Stavrakou et al., 2014) as suggested by field observations in Borneo (Langford et al., 2010), and (iii) the use of the crop distri-bution database of Ramankutty and Foley (1999) in MEGAN-MOHYCAN, along with the necessary adjustment of the other plant functional type distributions, leading overall to larger crop extent and lower total emissions.

The average global fire flux, expressed as burnt biomass, is reduced from 2028 TgC/yr (GFED4s) to 1653 TgC/yr after

optimization (Table 1). Note that the inversion provides updated VOC emissions of HCHO precursors. However, to ease the comparison with other inventories, VOC emissions are converted to carbon emissions through the use of emission factors obtained from the compilation of Andreae and Merlet (2001) (with 2011 updates). It should be acknowledged that the top-down estimates given here for fuel consumption might be affected by errors in the emission factors as well as on errors on the formaldehyde yields per VOC. The strongest emission decreases are induced over Africa (23%), South America and Southeast

Asia (15%), whereas in Europe the fire fluxes are 12% higher than in GFED4s. The reduced top-down emission agrees within 15% with the GFED4 inventory (1438 TgC/yr), and is ca. 18% lower than the GFAS and FINN global estimates. The lower a posteriori emissions in Africa are supported by the independent inventories, and the flux updates in Europe and Russia are in good agreement with the FINN fluxes. At tropical latitudes, the estimates from the independent inventories exhibit often large discrepancies, underscoring the large uncertainty of this source, while the top-down emissions lie generally within their range

(Table 1, Fig. 4).

The OMI-based fire emissions present a marked interannual variability, between a minimum of 1383 TgC in 2013 and a maximum of 1966 TgC in 2007 (Table 2). Figure 4 illustrates the coefficient of variability, defined as the standard deviation of the emissions divided by the mean, which is a measure of the interannual variability of the emissions (Giglio et al., 2013). The GFED4s coefficient is lowest over Africa (less than 0.15) and highest in South America, Southeast Asia, Russia and

Australia (0.35-0.57). The low variability over Africa can be explained by the dominance of intense savanna fires that are highly regular throughout the years. According to the source attribution of GFED4s, deforestation fires are by far the prevailing source responsible for 80% of the total emission in South America, while the rest is due to savanna burning occurring in northeastern South America. The coefficient of variation of South American deforestation fires amounts to 0.74, pointing to the strong effect of climate variability caused by e.g. the strong El Niño Southern Oscillation (ENSO) on the fire occurrence

in the Amazon (Alencar et al., 2011), and the rapid decline in deforestation rates over 2005-2013 (Nepstad et al., 2014). In addition, the estimated coefficient for savanna fires (0.41) is substantially higher than for the African savannas due to the strong variability of fire burning in the northern South America (Romero-Ruiz et al., 2010). In Australia, savanna, grassland, and shrubland fires are responsible for the high interannual variability of the GFED4s inventory (0.42). In Southeast Asia the contribution of peat burning to the total fire flux varies strongly from year to year (0-38%) and drives the high coefficient of

variability (0.45) (Giglio et al., 2013). After inversion, the coefficient of variability is reinforced over Europe and SH Africa,



but is reduced in the Tropics, especially over Southeast Asia and South America, where the decreased top-down variability is supported by comparisons with GFAS and FINN (Fig. 4).

The global mean 2005-2013 isoprene emission is reduced from 343 to 272 Tg/yr after inversion (Table 1), with the largest reductions inferred in NH Africa and South America (ca. 30%), and in Southeastern US (35%). In constrast to the emission

decrease suggested by satellite, the isoprene fluxes estimated by MEGAN-MACC and GUESS-ES are substantially higher, by 100% and 66%, respectively. The interannual variation of the isoprene fluxes is low in all regions, with the coefficient of variability close to 0.04 in the Tropics and up to 0.07 in extratropical regions. The satellite columns suggest stronger interannual variability over all regions, except in South America where it is slightly reduced. The interannual variation of isoprene fluxes is low for all inventories, generally stronger in MEGAN-MACC (up to 0.1) and weaker in GUESS-ES (Fig. 4).

The monthly variation of the a priori and the OMI-based emissions is compared directly to MODIS Aqua (MYD14CM, 13h30 LT) fire counts (http://reverb.echo.nasa.gov) over twelve smaller regions selected based on literature evidence for the occurrence of small fires (Table 3, Fig. S4). Higher spatial and temporal correlations are calculated after the inversion in all selected areas, especially over agricultural regions, like Southeastern US, Eastern Australia and Maranhão, where the correlation improves significantly, from 0.36 to 0.65, from 0.55 to 0.86, and from 0.56 to 0.91, respectively. This shows that satellite

HCHO observations do detect the contribution of small fires. It also explains the improved correlation of OMI-based emissions with GFAS and FINN in South America, Northern Africa and Southeast Asia, regions where this contribution is important (Chen et al., 2013; Huang et al., 2012; Karki, 2002; Magi et al., 2012).

The ratio of the optimized to the a priori annual fluxes for biomass burning and isoprene emissions is presented in Fig. 5 and Fig. 6, respectively. The interannual flux variation is displayed in Fig. 7 and Fig. 8 and the seasonal variation of the fluxes over

different regions (Sec. 4-8) are shown in Figs. 9, 11-15. We present detailed results for regions where the satellite observations suggest important changes relative to the a priori fluxes.

## 4   Amazonian emissions

The OMI columns suggest important fire flux decreases during years with strong a priori fluxes, by 16% in 2005, 22% in 2007 and 32% in 2010. The inferred flux reduction in 2010 is corroborated by earlier inversion studies constrained by GOME-2

HCHO columns (Stavrakou et al., 2015), MOPITT CO observations (Bloom et al., 2015), and a multi-sensor based emission estimate above Mato Grosso (Anderson et al., 2015). The top-down interannual fire variability is marked, but less pronounced compared to the a priori, with the lowest emission inferred in 2009 (80.2 TgC) and the highest in 2007 (387.4 TgC, Fig. 7), and is corroborated by the GFAS and FINN inventories (Fig. 4). The time and duration of the fire season is not modified by the optimization (Fig. 9). The OMI-derived fluxes display the same pronounced seasonality as GFED4s, with fire emissions peak-

ing between August and September, and a rapid decline in October and November, as found in previous studies (Barkley et al., 2008; Bloom et al., 2015; Stavrakou et al., 2015). The independent inventories, however, indicate generally higher fluxes than the top-down fluxes from October to January (Fig. 9).





Regarding isoprene, the inversion infers generally lower fluxes than the a priori inventory for all years of the study period, with a 38% mean annual reduction over 2005-2013, as illustrated in Fig. 6 and 8. The top-down annual isoprene flux ranges between 59 Tg in 2013 and 70 Tg in 2007 and the a priori interannual and seasonal variability is generally preserved after inversion (Fig. 8, 9), and similar in all inventories, with minimal emissions during the wet-to-dry season transition (April-June) and higher fluxes during the dry season (July-October). The peak-to-trough ratio is about a factor of 2 for the a priori and optimized fluxes, whereas it is weaker in GUESS-ES inventory (1.6) and stronger in MEGAN-MACC (2.4). During the wet-to-dry transition season (April-June), top-down estimates from GOME-2 and OMI show better consistency than in the dry season (Fig. 9, Stavrakou et al. (2015)). An all-year-round emission decrease in most bottom-up inventories was also required in order to reconcile the GEOS-Chem model with SCIAMACHY and OMI HCHO columns (Barkley et al., 2013). The strong seasonal variation and low emissions during the wet-to-dry transition are most likely due new leaf growth and lower flux rates from young leaves (Barkley et al., 2009).

Figure 10 shows a comparison of modelled isoprene fluxes with flux measurements from 12 field campaigns performed in the Amazon. The comparison accounts for the diurnal variations in the fluxes through correction factors used to scale the measured fluxes to daily averages (cf. Table S1). Direct comparisons between modelled fluxes and field measurements should, however, be considered with caution mainly due to the coarse resolution of the modelled emissions, but also to the fact that flux measurements were often performed outside the study period (2005-2013). The observed isoprene fluxes exhibit strong local differences within the forest (up to 5 mg/m$^2$/h, Karl et al. (2009)), as well as significant differences from one day to another (up to 0.5 mg/m$^2$/h) (Ciccioli et al., 2003; Karl et al., 2007; Kuhn et al., 2007)), whereas they might exhibit differences of up to 1 mg/m$^2$/h associated with the use of different measurement techniques (Helmig et al., 1998; Karl et al., 2007; Kuhn et al., 2007). Overall, the emission reduction inferred by the satellite observations lies within the variability of the field measurements, while the discrepancies between the observed fluxes are often larger than the differences between the a priori and a posteriori fluxes. The field studies generally agree on higher fluxes during the dry and the dry-to-wet transition season between July and December (Simon et al., 2005), while a recent field campaign suggests much lower fluxes (by ca. factor of 3) compared to the top-down estimates, most likely related to a local effect of leaf flushing at the measurement location (Alves et al., 2015).

# 5   African emissions

In Northern Africa, the biomass burning source is reduced by the inversion by 15-38% for the different years and lies closer to GFED4, GFAS and FINN estimates (Table 1, Fig. 11). In this region, both natural and agricultural fires peak in December, while the agricultural fires are dominant in the beginning, from September to November, and at the end of the fire season, from February to April (Magi et al., 2012). The OMI observations suggest ca. 50% emission decrease in the fire peak season, which is supported by comparisons with GFAS and FINN inventories (Fig. 11), and a moderate increase from February to April when the agricultural fires are dominant, and when the fraction of small fires is largest according to GFED4s. Higher emissions from February to April are also supported by GFAS and FINN, suggesting an even stronger shift in the fire season,





with higher fire emissions lasting until May. The reduced emission amplitude and the longer burning season in Northern Africa are corroborated by an inversion study using CO columns from the MOPITT instrument (Chevallier et al., 2009).

In Africa south of the equator, the OMI-based fire source is 23% lower than the bottom-up estimate, and lies closer to the estimates of GFED4, GFAS and FINN (Table 1, Fig. 11). In terms of seasonal variation, the natural fires open the fire season

between April and October, followed by agricultural fires lasting from June to November (Magi et al., 2012). The inversion infers 21% lower emissions in the beginning of the fire season, when fires are predominantly natural, a reduction by 43% during the fire peak between July and September, and by 20% higher emissions than GFED4s in October, when agricultural fires are the prevalent source (Fig. 11). The GFED4s inventory allocates the maximum of small fires fraction at the peak of the fire season (Randerson et al., 2012), resulting in an enhanced emission peak in July-August, rather than in September, as suggested

by the OMI observations. This seasonality shift of the burning season was also reported in past inversion studies constrained by SCIAMACHY and GOME-2 HCHO (Stavrakou et al., 2009b, 2015) and MOPITT CO observations (Chevallier et al., 2009).

Southern Hemisphere Africa can be divided in two regions based on the fire source updates suggested by OMI (Fig. 5). In its northern part, reduced emissions are systematically derived for all years, by up to 65%, with regard to the a priori flux, whereas in its southern part, (Southern Africa in Fig. 3), the emissions exhibit a stronger variability, increasing significantly until 2010,

but remaining closer to the a priori in the subsequent years, as illustrated in Fig. 5. The a posteriori emissions during the peak fire season in September are found to be up to a factor of 3 higher than FINN, and by 50% higher than GFAS and GFED4. The largest top-down flux in this region is inferred in September 2008, estimated at 50% higher than the a priori, due to record-high wildfires in Mozambique, South Africa and Swaziland in that year (Jha, 2010).

The OMI observations suggest a decrease of isoprene fluxes over the African continent by ca. 20% for all years of the target

period, from 79 Tg/yr in the a priori to 63 Tg/yr, as shown in Table 1. This decrease is very similar to the result obtained from an inversion study constrained by the NASA OMI HCHO retrieval product reporting an emission reduction in African isoprene fluxes, from 87 Tg/yr in the a priori to 68 Tg/yr through 2005-2009 (Marais et al., 2012). In the latter study, the flux decrease was strongest over equatorial and Northern Africa, in very good agreement with the updates shown in Fig. 6. In a follow-up inversion study also based on OMI observations, Marais et al. (2014) invoked a reduction of MEGAN emission

factors for broadleaf trees and shrub (ca. factor of 2) and woody savannas (20%) in Africa in order to reconcile the model with the observations, whereas the reported comparisons with ground-based measurements suggested that even lower isoprene flux rates might be necessary.

In Northern Africa, the isoprene fluxes in Northern Africa exhibit a weak interannual variability (Fig. 8, 11). The OMI observations point to a mean (2005-2013) decrease of 26% in this region with respect to the bottom-up estimate. The geo-

graphical extent of the emission updates (Fig. 6) is in agreement with previous satellite-based results using SCIAMACHY (Stavrakou et al., 2009b) and GOME-2 HCHO columns (Bauwens et al., 2014; Stavrakou et al., 2015). As seen in Fig. 11, the seasonality of isoprene emissions in Northern Africa is characterized by two emission maxima, driven by the two equatorial rainy seasons occurring from March to May and from August to November. The satellite columns indicate a change in the seasonal profile, from two equally strong emission maxima in April-May and in October, to a peak in March and a weaker second

peak in October-November (Fig. 11). This agrees with the seasonality derived from GOME-2 observations, and is similar to





the seasonality change reported by Marais et al. (2012). The stronger emissions in the first half of the year are also consistent with the independent inventories, whereas the secondary peak is better represented in GUESS-ES inventory.

The isoprene emissions in Southern Africa peak during the Southern Hemisphere summer, when both temperature and precipitation rates are higher (Fig. 11). Both MEGAN-MACC and GUESS-ES emission estimates are about a factor of 2 higher than the top-down estimates. The discrepancy with MEGAN-MACC is partly explained by the neglect of the soil moisture stress effect ($\gamma_{SM}$) in the standard version of the MEGAN-MACC model. Its inclusion in MEGAN-MACC was found to have a strong impact, leading to a flux decrease by 50% on global scale, and even stronger decreases in Africa and South America (Sindelarova et al., 2014). Interestingly, the inversion suggests a large increase of isoprene emissions (up to a factor of 2) southward of 15° S, and particularly in the very dry Southwestern part of the continent (west of ca. 30° E), where the soil moisture stress effect is strongest in the MEGAN-MOHYCAN emissions (Fig. 6 and Fig. 2 in Müller et al. (2008)). The spatial coincidence of the largest emission updates inferred by the inversion with the areas where the soil moisture stress effect is strongest is a first indication that its parameterization in MEGAN overestimates the impact of very low soil moisture on the emissions in dry subtropical environments like Southern Africa (also Australia, see Sec. 7). A second, even stronger indication is provided by the interannual variability of the emission updates in Southwestern Africa (15–35 S, 10–30 E) shown on Fig.12. These updates are indeed found to be well correlated ($r = 0.81$) temporally with the factor by which the emissions are reduced due to the soil moisture activity factor $\gamma_{SM}$. In other words, the emission increments are largest when and where $\gamma_{SM}$ is lowest.

MEGAN simulates isoprene response to soil moisture stress with a simple parameterization that shuts off isoprene emission when soil moisture drops to the level where plants can no longer draw moisture from the soil, known as the wilting point. While the MEGAN soil moisture stress effect uses a simple concept, the implementation is difficult due to the need to accurately model soil moisture, soil wilting point, and plant rooting depth. Seco et al. (2015) evaluated the MEGAN response to soil moisture stress by comparison to measured whole canopy isoprene fluxes and found that the algorithm performed poorly with the default soil wilting point but worked well when a more accurate value was used.

## 6 Emissions in Southeast Asia

The fire season in Southeast Asia is characterized by a first peak in March, associated with aboveground vegetation burning in former Indochina, and a second peak in August to October caused by peat combustion occurring in Indonesia (Chang and Song, 2010) (Fig. 13). The GFED4s fluxes vary considerably across the years, ranging between a minimum of 123 TgC (in 2011) and 277 TgC (in 2006). The top-down estimates remain generally close to the a priori, except in 2006 and 2009 where the satellite observations suggest a significant decrease of the fluxes associated to peat burning in Indonesia (Reddington et al., 2014) by almost factor of 3 (Fig. 13). The optimized fluxes generally increase in March and decrease from August to October, while the amplitude of the seasonal pattern is reduced, with the emissions in March being generally larger than the peat burning emissions in August. In addition, the higher a posteriori correlation with monthly MODIS fire counts in Indochina (Table 3)



indicates an improved representation of the seasonal natural fires in March-April and agricultural waste burning in April-May (Magi et al., 2012).

In Indonesia, the fire season extends from June to November, and comprises intense peat burning, in particular during extreme drought conditions caused by El Niño (Schultz et al., 2008; Worden et al., 2013). The GFED4s estimates are generally lower than 100 TgC/yr, but significantly higher for El Niño years, e.g. 2006 (350.3 TgC) and 2009 (191.6 Tg/yr). The inferred flux drop in 2006 and 2009 is supported by GFAS and FINN, but in all other years both FINN and GFAS are relatively close to GFED4s. The lower 2006 flux suggested by the observed columns is corroborated by an independent carbon emission estimate based on burned area in a small region of Borneo in 2006 (Central Kalimantan, approximately 13% of the Indonesian peatland) reporting peat fire emissions of 49 TgC during the 2006 El Niño episode (Ballhorn et al., 2009). This estimate is about half of the GFED4s value (109 TgC), and closer to the OMI-based estimate of 33 TgC for the same area and year. Note, however, that this independent estimate does not account for above ground biomass burning.

As mentioned in the previous sections, the updated isoprene emissions are systematically decreased in tropical regions, by about 40% on average in Amazonia and equatorial Africa (Fig. 6), pointing to potentially overestimated emission factors used in the MEGAN model for tropical forests. In contrast to these regions, the emission reduction for the tropical rainforests of Southeast Asia is much weaker (<20%, Fig. 6, 8) due to the lower basal emission rates incorporated in MEGAN-MOHYCAN (Stavrakou et al., 2014) based on OP3 campaign measurements in the rainforest of Borneo (Langford et al., 2010). The relatively small discrepancy between the model and the satellite HCHO columns in Southeast Asia supports the use of lower isoprene flux rates for the Asian rainforests.

In China, most of the fires are agricultural and their emissions are generally low, except for the North China Plain (Fig. 5, Stavrakou et al. (2016)). The isoprene fluxes in China are also reduced after optimization, from 7.3 Tg/yr in MEGAN-MOHYCAN to 5.8 Tg/yr on average over the study period, but the decrease is stronger in South China, ranging between 27% and 45% depending on the year. The emissions peak in summertime and present weak interannual variability (Fig. 8), with a maximum in 2007 (2.6 Tg/yr) and a minimum in 2010 (1.7 Tg/yr, Fig. 13). The OMI-based flux in 2010 is in good agreement with an earlier estimate inferred from GOME-2 HCHO observations (2.4 Tg/yr, Fig. 13) (Stavrakou et al., 2015).

## 7 Australian emissions

Northern Australia is a major fire-prone area where bushfires occur during many months every year (Steffen et al., 2015). The peak of the fire season is observed between September and November, but its magnitude depends strongly on the year. The fire season sets off between April and June, with the beginning of the dry season, gets reinforced by the hot temperatures and winds of the subsequent months, and lasts until December. The OMI data suggest top-down fluxes close to the a priori in all years, except for 2011, where the emission maximum is decreased by about 25% with respect to GFED4s (Fig. 14), whereas the estimates from GFAS and FINN in this region differ by more than a factor of 10. In Southern Australia (Fig. 14), the fire fluxes are generally half those in Northern Australia, and bushfires are again the main fire type in this region. This region, and in particular the state of Victoria, sometimes experiences extreme fire events, like the 2006-2007 bushfires which was one of





the worst in record, and the "Black Saturday" bushfires in February 2009. The satellite columns of HCHO lead to a significant reduction (Fig. 14) of the fire emission during the aforementioned major fire events in comparison to the GFED4s inventory, in good agreement with the FINN estimates.

The optimization indicates negative isoprene updates in the tropical and subtropical ecosystems of Northern Australia, which are dominated by woodland and grasslands, and generally positive flux increments in the southern part of the continent, where temperate forests and grasslands are prevalent (Fig. 8, 14). The mean reduction over 2005-2013 in Northern Australia amounts to ca. 20% with respect to the a priori (24.4 Tg/yr), and is supported by the inversion study based on GOME-2 HCHO columns (Stavrakou et al., 2015) as shown in Fig. 8, pointing to possibly overestimated emission factors assumed in MEGAN for tropical ecosystems. In Southern Australia, the a posteriori isoprene fluxes are increased by about 20% on average over the study period, from 12.5 Tg/yr to 15 Tg/yr, and show small interannual and seasonal variability (Fig. 14). Although the MEGAN-MACC emissions are much higher than the other inventories over Australia, a sensitivity calculation accounting for the soil moisture stress activity factor in the MEGAN-MACC model resulted in a substantial flux decrease of about 70% with respect to the reference MEGAN-MACC simulation (Sindelarova et al., 2014), stressing the important role of soil moisture stress in these very dry environments. As for Southern Africa, the OMI-based inversion over Southern Australia enhances the emissions where and when $\gamma_{SM}$ reaches its lowest values (Fig. 6 and 12). As discussed above, the poor performance of the parameterization could be partly due to misrepresentations of driving variables (soil moisture content) or soil characteristics (wilting point, rooting depth). The use of satellite-derived soil moisture or solar-induced fluorescence (van der Molen et al., 2016; Joiner et al., 2016) could be a promising way for improving the soil stress estimation in the future.

## 8  Mid-latitude emissions

In Europe, the fire season peaks in summertime and a secondary peak is also recorded in spring, mainly due to emissions from agricultural waste burning (Fig. 15). The optimized fluxes lie generally close to the a priori except in 2006 and 2007 where the OMI observations point to higher fluxes (by 40-50%) than in GFED4s during the emission peak. The strong flux in April-May 2006 and in summer 2007 were due to numerous agricultural fires that occurred in the Baltic countries, western Russia, Belarus, and Ukraine (Stohl et al., 2007), and to intense biomass burning in southern Europe. The increase in the top-down estimates in 2007 is in line with the reported increase based on IASI CO columns (Turquety et al., 2009). The top-down estimate agrees well with GFED4s during the devastating fires in the Moscow area in July-August 2010, whereas previous studies reported values which were a factor of 2 (Yurganov et al., 2010), 3 (Konovalov et al., 2011), or 10 (Krol et al., 2013) higher than the older GFED3 inventory (van der Werf et al., 2010), which was by about 60% lower than GFED4s in this region.

Regarding isoprene fluxes over Europe, the satellite observations suggest an average increase by 15% in Western Europe (from 2.9 to 3.4 Tg/yr), and by 33% in Eastern Europe (from 3.9 to 5.2 Tg/yr), whereas the inferred increase is significantly stronger during extremely hot summers, like in 2007 and 2010. Indeed, in July 2007 Greece experienced the hottest summer on record since 1891 (Founda and Giannakopoulos, 2009) with temperature anomalies of +5°C compared to the 1961-1990 mean, and in July 2010, the hottest summer since 1500 was recorded in western Russia with temperature anomalies of +6°C with



respect to the 1961-1990 mean (Barriopedro et al., 2011; Coumou and Rahmstorf, 2012) (http://www.ncdc.noaa.gov/temp-and-precip).

The concurrence of pyrogenic and isoprene emissions in the mid- and high-latitudes of the Northern Hemisphere during summertime is, however, an inherent difficulty in the derivation of top-down emissions by inverting for HCHO columns.
HCHO being an intermediate compound in the oxidation of both pyrogenic and biogenic hydrocarbons, it cannot be excluded that the HCHO column enhancements associated to higher isoprene emissions, have in reality a pyrogenic origin and vice versa. The inversion scheme relies strongly on the a priori emission distributions and errors in the retrievals, and thereby, errors in the geolocation of fire hot spots in the bottom-up inventories could propagate as errors in the source attribution, in particular for intense fire events associated to summer heat waves.

In Southeastern US, a major isoprene emitting region, the top-down fluxes are systematically reduced compared to the initial inventory, by 35% on average, with the strongest decrease (50%) inferred in 2013. Similarly to the a priori, the a posteriori estimates peak in 2011 and are lowest in 2013. This variability is primarily related to temperature changes, with recorded temperature anomalies of +3°C in 2011, and -1.5°C in 2013, with respect to the 1961-1990 mean (http://www.ncdc.noaa.gov/temp-and-precip/). The use of GOME-2 HCHO columns to constrain the inversion in 2010 (Stavrakou et al., 2015) results in an excellent agreement with the OMI-based fluxes (Fig. 8 and 15), whereas both optimizations suggest a slightly modified seasonal profile, with a primary peak in June and a secondary in August. The need for lower emissions in Southeastern US compared to the MEGAN model has been put forward by past studies based on satellite observations of HCHO from GOME, SCIAMACHY and OMI sensors (Palmer et al., 2006; Millet et al., 2008; Stavrakou et al., 2009b). The MEGAN-MACC and GUESS-ES estimates are in very good agreement with the a posteriori fluxes in terms of magnitude, although in some years the peak emission is delayed by one month (Fig. 15).

## 9 Emission trends

The global distribution of isoprene emission trends over 2005-2013 according to the bottom-up emission inventory and as suggested by the inversion of satellite data is displayed in Fig. 16. Although deriving long-term emission trends from satellite data might be very useful for diagnosing global and regional change, particular caution is required when interpreting the results, since physical changes in the satellite instruments over time might result in artificial drifts in the observations. In the case of OMI HCHO columns, special efforts were made to reduce the effects of the row anomaly issue (http://projects.knmi.nl/omi/research/product/rowanomaly-background.php) on the retrieved HCHO columns, in order to ensure the suitability of the data for addressing trend studies (De Smedt et al., 2015). Nevertheless, it appears difficult to avoid that any time-dependent instrumental effect might impact the interannual variability of emissions reported in this section.

Amazonia experienced a rapid decline of pyrogenic emissions, estimated at -7%/yr in the a priori GFED4s inventory and -8%/yr in the OMI-based emissions as a result of the trend in OMI columns calculated during the dry season (-3.2%/yr in August-September). This trend in HCHO columns was attributed to a strong decline in deforestation rates in the Amazon (Nepstad et al., 2014), and especially in Mato grosso and Rondônia where the cover loss in evergreen broadleaf forests de-



creased by more than 80% between 2002 and 2009 (Fanin and van der Werf, 2015). The isoprene emission trend over Amazonia, which is close to negligible (-0.2%/yr) in the a priori inventory (Fig. 9 and 16) becomes negative after optimization (-2.1%/yr). The derivation of biogenic emission trends in this region is made difficult by the magnitude and strong interannual variability of biomass burning. However, a decline of isoprene emissions is supported by the negative trend (-1.3%/yr) in the observed HCHO columns during the wet season (November-April), when biomass burning plays only a very minor role. This result is difficult to interpret. Recent findings based on satellite surface reflectance data (more precisely, normalized difference vegetation index, or NDVI, data) point to diminished vegetation greenness since 2000 due to a precipitation decline across large parts of Amazonia, especially Northern Brazil (Hilker et al., 2014). However, most of these changes occurred between 2000 and 2005, whereas the precipitation rates and NDVI values were comparatively more stable afterwards, and the Leaf Area Index from MODIS Collection 5 (MOD15A2 composite, http://modis.gsfc.nasa.gov/data/dataprod) either increased (by less than 0.5%/yr) or showed no trend over 2005–2013 over most of Amazonia (see Fig. S5).

Over Northern Africa, during the fire season (November-February) a decreasing trend of about 3%/yr over the study period is derived for the OMI-based fire fluxes (Fig. 7), close to the GFED4s trend (-3.2%/yr), whereas the corresponding trend of FINN (-2.4%/yr) is somewhat weaker. This trend is most likely related to negative trends observed in burned area in Northern Africa (Andela and van der Werf, 2014; Giglio et al., 2013) owing to land use changes (conversion of savannah into cropland), and to changes in precipitation, driven by the El Niño/Southern Oscillation (Andela and van der Werf, 2014).

In Siberia, the strongly positive isoprene emission trend of the bottom-up inventory (3.8%/yr) (Fig. 16) is a result of the warming temperature trends in this region (0.12°C/yr over 55-75 N, 40-120 E, based on ECMWF ERA-Interim temperature data over 2005-2013). The model incorporates both the direct effect of warming on the MEGAN temperature response of the emissions and the indirect effect through the increase in LAI, which reaches 3%/yr in Northern Siberia (Fig. S5). The inversion leads to an even higher trend (4.2%/yr), induced by the strongly positive trend in the HCHO observations (4.2%/yr) over 2005-2013 in this region. Higher temperatures might also favor the extension of forests inducing even higher isoprene emissions (Potosnak et al., 2013). According to MODIS land cover data, the forest fraction in this region has increased from 31% in 2005 to 36% in 2012 (Friedl et al., 2010).

Opposite a priori isoprene trends are calculated in Western and Eastern Europe over the study period, -2.5%/yr and 3.2%/yr, respectively, mostly related to the temperature and solar radiation trends. The OMI observations corroborate these trends, showing a negative trend in Western Europe (-1.1%/yr) and a positive trend in Eastern Europe (0.4%/yr). The calculated trends after optimization are moderately enhanced, -3.3%/yr and 3.9%/yr in Western and Eastern Europe, respectively. Besides climate parameters, land use changes might also contribute to the increasing column and emission trend in Eastern Europe. Based on MODIS land cover data (Friedl et al., 2010), the forest fraction increased at a faster pace in Eastern than in Western Europe (1.1%.yr vs. 0.9%yr), and the crop fraction decreased more rapidly (-0.5%/yr) in Eastern than in Western Europe (-0.4%/yr).

Over the Southeastern US, the slightly negative trend in the summertime isoprene fluxes in the a priori (-0.3%/yr) becomes much more pronounced after inversion (-4%/yr), induced by the downward trend in the OMI HCHO columns (-2.5%/yr) over 2005-2013 (De Smedt et al., 2015). Except for this trend, the interannual variability of the top-down emissions over this region is similar to the a priori (Fig. 8). The long-term decline could be in part an artefact resulting from the well-documented





downward trend in NOx abundances over the United States (Russel et al., 2012; Kharol et al., 2015), which could significantly decrease formaldehyde production over time, if the yield of HCHO per isoprene molecule is substantially lower at low NOx level than at high NOx. The ground-level $NO_2$ concentrations have decreased by as much as a factor of two over Eastern US based on OMI and in situ measurements between 2005 and 2012 (Kharol et al., 2015). The NOx dependence of the HCHO

yield is taken into account in the calculations presented in this study, but the modelled decrease in PBL NOx level in the Eastern US is lower (ca. -30%) than observed (-50%) during 2005–2012. Furthermore, the low-NOx oxidation mechanism remains incompletely characterized, especially regarding the further degradation of primary oxidation products, leaving open the possibility of a significant overestimation of the HCHO yield at low NOx, even though a recent analysis of airborne measurements over the Southeast US indicated that state-of-the-art oxidation mechanisms can reproduce the NOx dependence

of prompt HCHO formation inferred from the measurements (Wolfe et al., 2016). If confirmed, an overestimation of the HCHO yield at low NOx could also help to explain the negative trend in top-down isoprene emission over Western Europe (Fig. 16). More importantly, it would imply a general underestimation of our top-down emissions in low-NOx environments, tropical forests in particular.

In South China, the negative summertime trend (-0.7%/yr) in HCHO columns drives a change in the sign of the 2005-2013

isoprene trend, from 0.1%/yr in the a priori to -1.6%/yr in the OMI-based fluxes. The very small a priori emission trend results from a combination of compensating effects : on one hand, declining trends in the ERA-Interim photochemically active radiation (PAR) (-0.33%/yr) and temperature (-0.03 K/yr), and on the other hand, an increasing trend in leaf area index (1%/yr, cf. Fig S5) and a decline in crop extent in South China, suggested by the land use database of Ramankutty and Foley (1999) and supported by MODIS land cover data (Friedl et al., 2010). However, a recent land cover database suggests that the extent

of crops has increased in eastern China in the last 30 years (Hurtt et al., 2011). In addition, the declining trend in PAR was also derived from ERA-Interim data complemented by surface radiation measurements (Weedon et al., 2014). The crop expansion and declining PAR were proposed to cause a negative isoprene trend in South China (Yue et al., 2015), and likely explain the observed negative trend in HCHO.

## 10   Conclusions

Global distributions of pyrogenic and biogenic VOC fluxes between 2005 and 2013 were derived using the adjoint inversion scheme built on the IMAGESv2 global CTM and HCHO column abundances retrieved from the OMI sounder. The inversion suggests a moderate decrease (ca. 20%) of the global average emissions of both pyrogenic and and biogenic VOCs relative to the a priori emissions used in the model. The main findings of this study are presented below.

  – The global top-down fire fluxes exhibit strong interannual variability, ranging between ca. 1400 TgC/yr (2011) and 2000

TgC/yr (2007). The a priori interannual variability is generally well preserved, but the inferred estimates are by ca. 250 to
      450 TgC lower than the a priori depending on the year, with the largest decreases suggested over Africa, South America,
      and Southeast Asia (23%). The top-down emissions are better correlated with MODIS fire counts in regions with small



fires than GFED4s, indicating that the associated emissions might be too low in GFED4s, and that they can be derived by the OMI-based inversion.

– The inversion suggests (i) important fire flux decreases (15-30%) in Amazonia during years with strong a priori emissions, (ii) about 50% emission decrease during the peak fire season in Northern and Southern Africa, (iii) generally increased emissions in Indochina and decreased fluxes in Indonesia during intense fire events related to El Niño years, (iv) significant flux reduction during the major bushfires in Australia, and (v) flux increases during the devastating fires in Southern Europe in 2007.

– Changes in fire seasonal patterns are suggested, in particular in Southeast Asia and Africa. In Southeast Asia, the seasonal amplitude is reduced after inversion, with enhanced emissions due to aboveground vegetation burning in March, and weaker emissions due to Indonesian peat burning in August. The inversion suggests generally increased fluxes due to agricultural fires over Africa, and decreased emission maxima due to natural fires.

– Significant reductions of isoprene fluxes are inferred in tropical ecosystems (30-40% in Amazonia and Northern Africa), suggesting overestimated basal emission rates in these areas. The top-down fluxes generally increase over Eurasia, especially during heat waves in summer (e.g. western Russia in 2010), suggesting a possibly stronger emission response to high temperatures than currently assumed.

– The inversion suggests large isoprene emission increases (up to 100% locally) over areas most affected by the soil moisture stress parameterization in MEGAN, in particular in Southern Africa and Southern Australia. The inferred isoprene increments present a strong interannual correlation with $1/\gamma_{SM}$, i.e. by the factor by which isoprene emissions are reduced due to soil moisture stress in MEGAN (r$\geq$0.7), indicating that the soil moisture parameterization leads to overly decreased isoprene fluxes.

– The isoprene emission trends are found to be often enhanced after inversion. Positive trends in top-down isoprene emissions are inferred in Siberia (4.2%/yr) and Eastern Europe (3.3%/yr), likely reflecting forest expansion and the warming trend. Negative trends are derived in Amazonia (-2.1%/yr), South China (-1%/yr), the United States (-3.7%/yr), and Western Europe (-3.9%/yr). The top-down trends should be considered with caution due to possible drifts in the satellite data. In several instances, however, they are supported by independent evidence from literature studies. Trends in NOx emissions might play a significant role given their possibly large influence on formaldehyde yields, which remain imperfectly characterized and deserve more attention, especially at low NOx.

For simplicity and to avoid excessive computational costs, a detailed error assessment of the a posteriori emission estimates is not addressed in this work. Nevertheless, sensitivity inversions conducted in an earlier study, also based on OMI columns for 2010, have shown that the inferred fluxes were generally weakly dependent on the choice of key model and inversion parameters, and lay within 7% of the standard inversion results (Stavrakou et al., 2015). Recent developments in the representation of vertical profiles of smoke released by open fires (Sofiev et al., 2013), in the partitioning of burned biomass into emitted





trace gases (Akagi et al., 2011), and in the spatiotemporal variability of emission factors (van Leeuwen et al., 2011, 2013) are additional sources of uncertainty that could impact the top-down fluxes and should therefore be carefully assessed in future studies.

*Acknowledgements.* This research was supported by the Belgian Science Policy Office through the PRODEX projects ACROSAT, by the European Space Agency (ESA) through the GlobEmission DUE project (2011-2016), and by the MARCOPOLO project (2014-2016) funded by the European Commission within the Seventh Framework Programme (grant agreement : 606593).





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





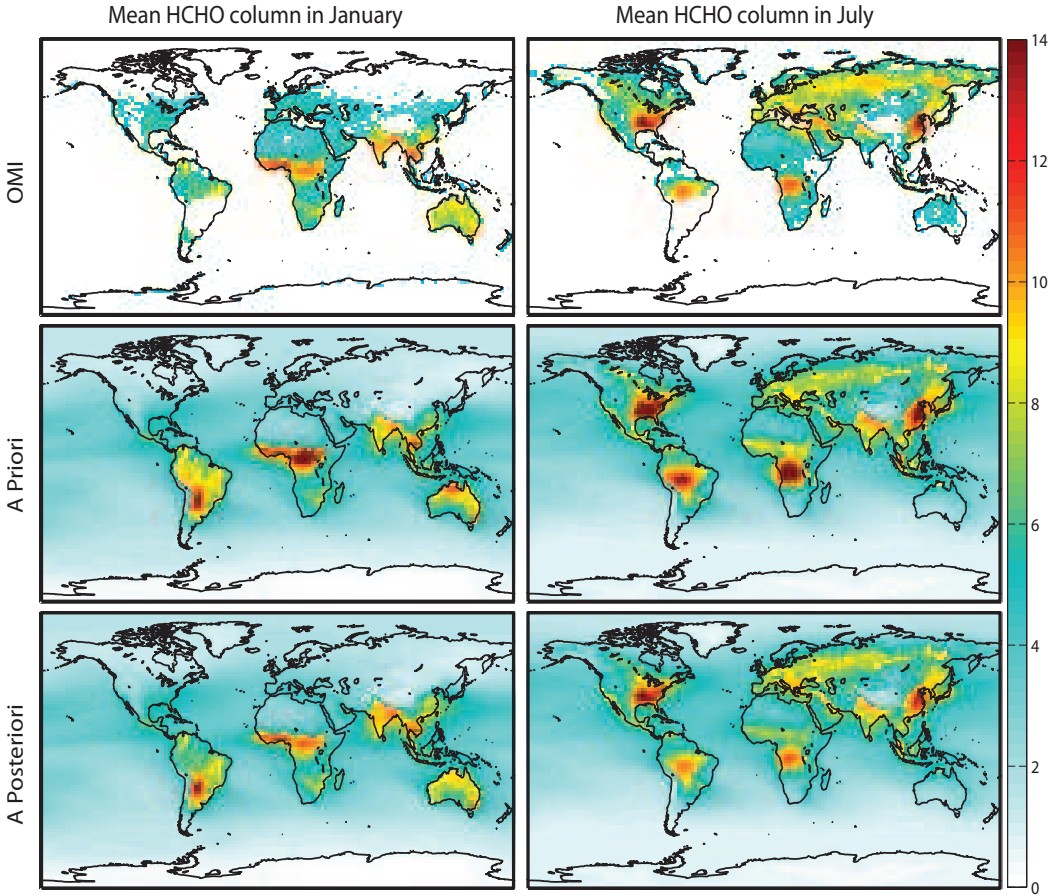

**Figure 1.** Global distributions of mean 2005-2013 HCHO columns for January and June observed by OMI (upper panels), modelled using emissions (middle panels) and inferred after optimization (lower panels). The columns are expressed in $10^{15}$ molec.cm$^{-2}$. The observed monthly averages exclude scenes with cloud fractions higher than 40% and land fractions lower than 20%, as well as data with a retrieval error higher than 100%.



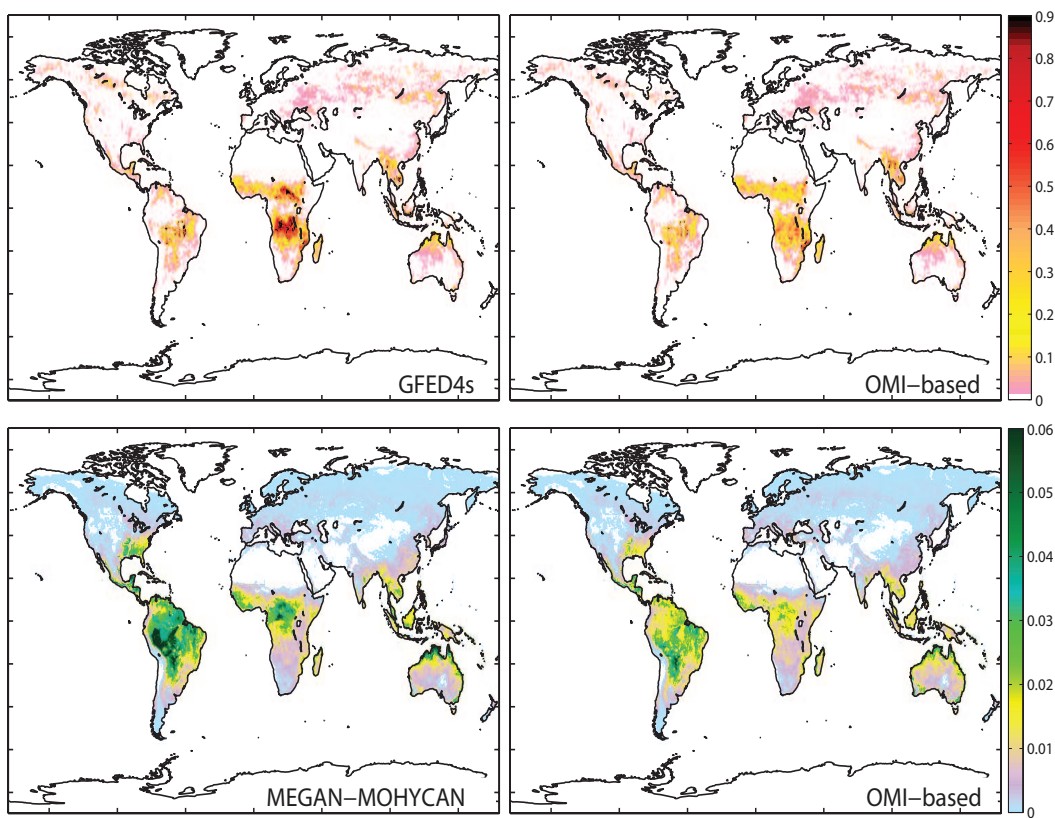

**Figure 2.** Upper panel: Mean (2005-2013) annual biomass burning emission estimates in TgC/grid per year according to the a priori inventory GFED4s and to the OMI-based biomass burning emissions. Lower panel: Mean (2005-2013) annual isoprene emission estimates in Tg isoprene per grid cell per year according to the a priori MEGAN-MOHYCAN inventory and from the OMI-based inversion.





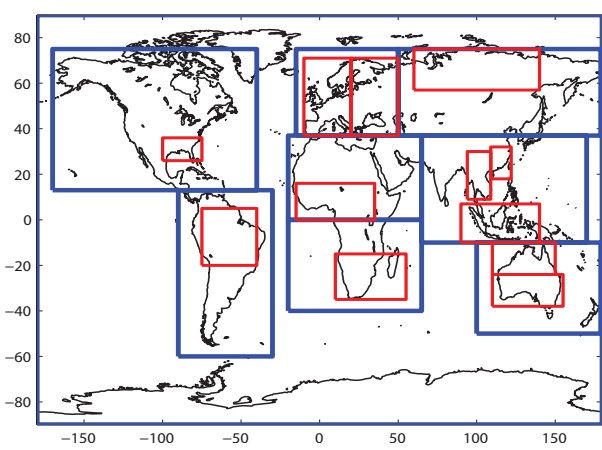

**Figure 3.** Definition of big and small regions used in this study. Big regions are : N. America (13-75 N, 40-170 W), S. America (60 S-13 N, 90 W-30 E), Europe (37-75 N, 15 W-50 E), NH Africa (0-37 N, 20 W-65 E), SH Africa (0-40 S, 20 W-65 E), Russia (37-75 N, 50-179 E), SE Asia (10 S-37 N, 65-170 E), Australia (10-50 S, 110-179 E). Small regions are : SE US (26-36 N, 75-100 W), Amazonia (5-20 S, 40-75 W), W. Europe (37-71 N, 10 W-20 E), E. Europe (37-71 N, 20-50 E), Northern Africa (0-16 N, 15 W-35 E), Southern Africa (15-35 S, 10-55 E), Siberia (57-75 N, 60-140 E), South China (18-32 N, 109-122 E), Indochina (9-30 N, 94-109 E), Indonesia (10 S-7 N, 90-140 E), N. Australia (10-24 S, 110-150 E), S. Australia (24-38 S, 110-155 E).



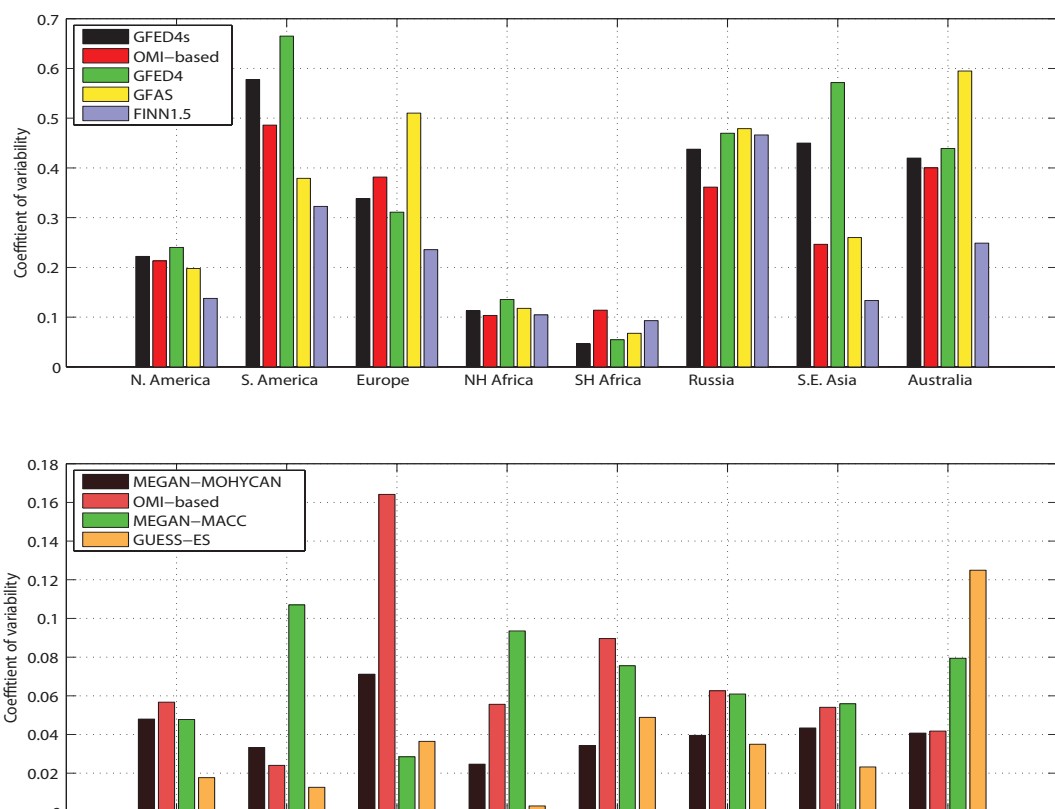

**Figure 4.** Interannual variability expressed as coefficient of variation, defined as the standard deviation of the emissions divided by the mean of the emissions, given for the a priori, for the OMI-based emission estimates and for the independent emission inventories of biomass burning (upper pannel) and isoprene emissions (lower panel) over the big regions defined in Fig. 3.





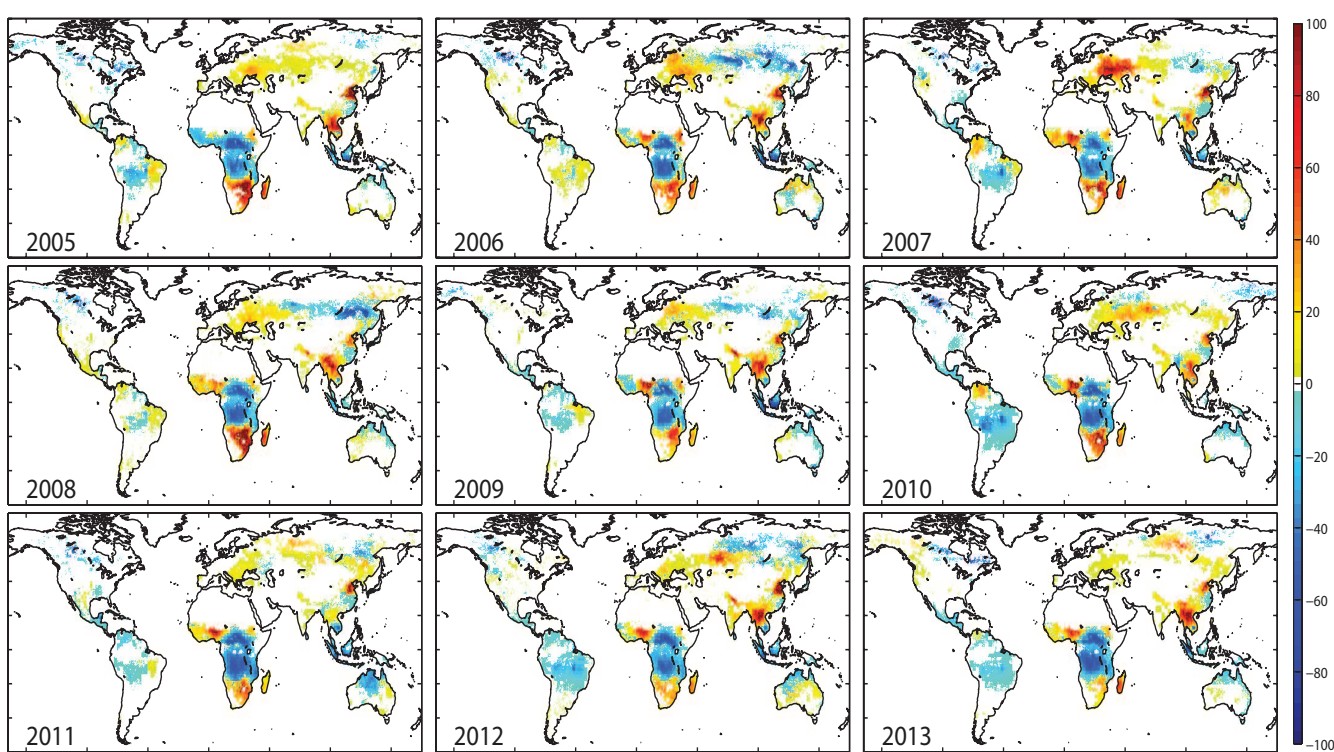

**Figure 5.** Updates (percentage change from the a priori) in annually averaged biomass burning emissions suggested by the flux inversion for all years of the study period.



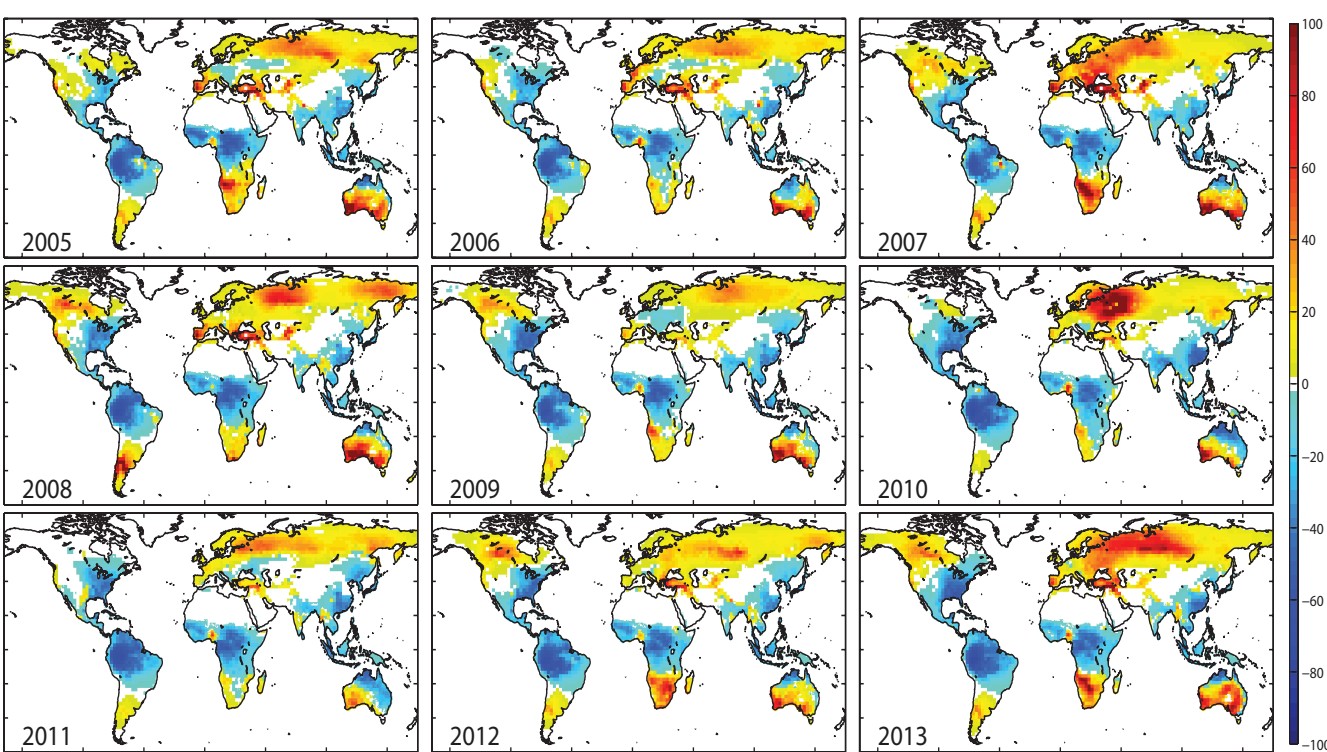

**Figure 6.** Updates (percentage change from the a priori) in annually averaged isoprene emissions inferred by the optimization for all years of the study period.





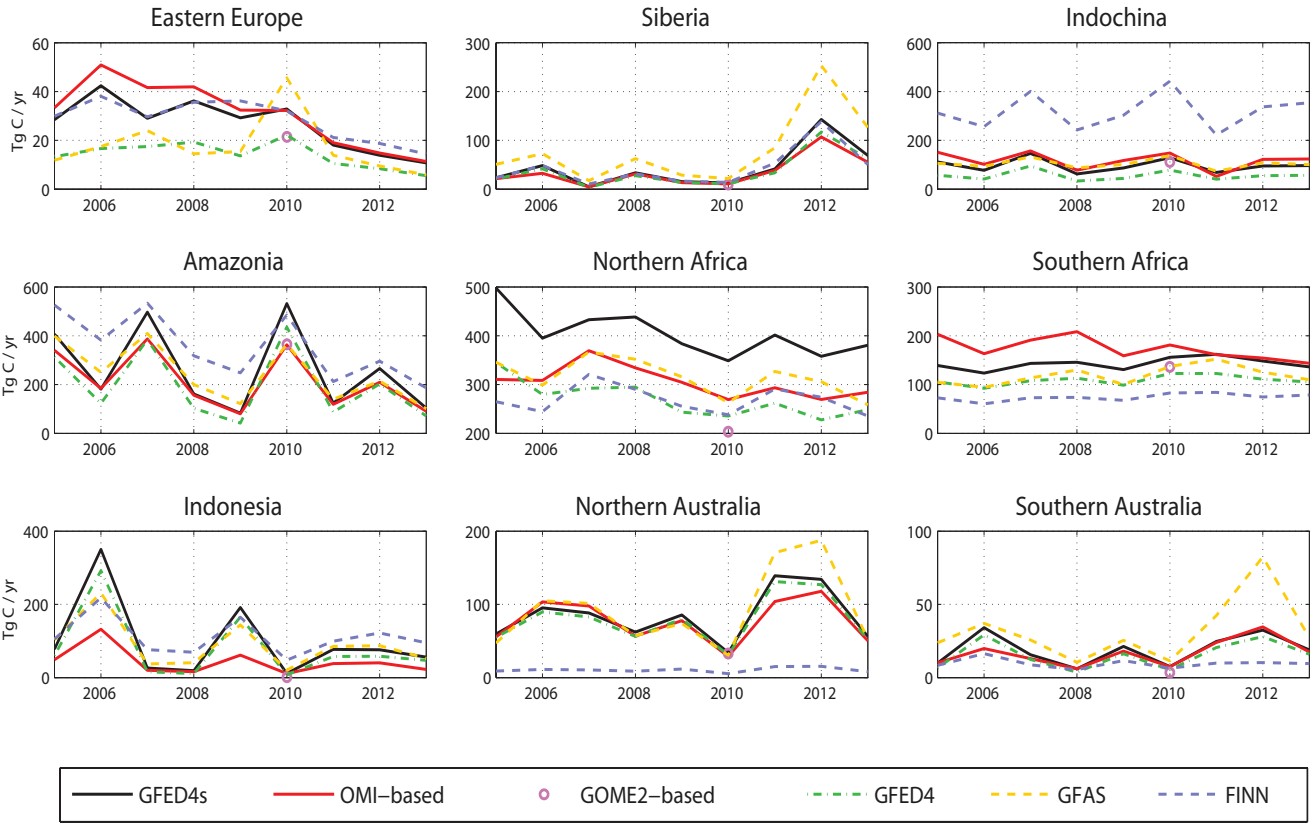

**Figure 7.** Interannual variation of burnt biomass (in TgC/yr) over 2005-2013 from the a priori inventory (black), the satellite-based estimates (OMI in red), and from other bottom-up inventories (GFED4 in green, GFAS in orange, FINN in blue) over small regions defined in Fig. 3. Units are TgC/yr. The GOME-2-inferred estimate is shown as magenta circle.




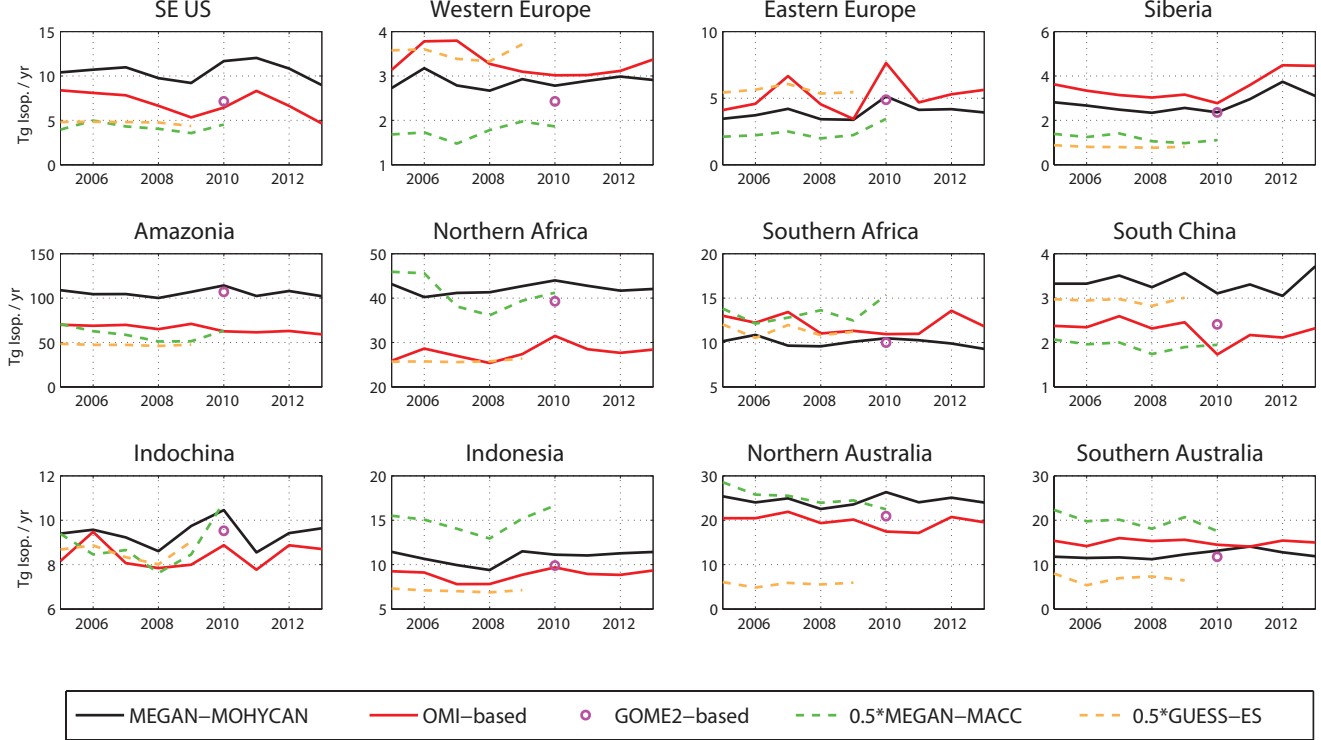

**Figure 8.** Interannual variation of isoprene fluxes over 2005-2013 from the a priori inventory (black), the satellite-based estimates (OMI in red), MEGAN-MACC (in green) and GUESS-ES (in orange) over regions (red boxes) defined in Fig. 3. Units are Tg of isoprene per month. The GOME-2-inferred estimate is shown as magenta circle.





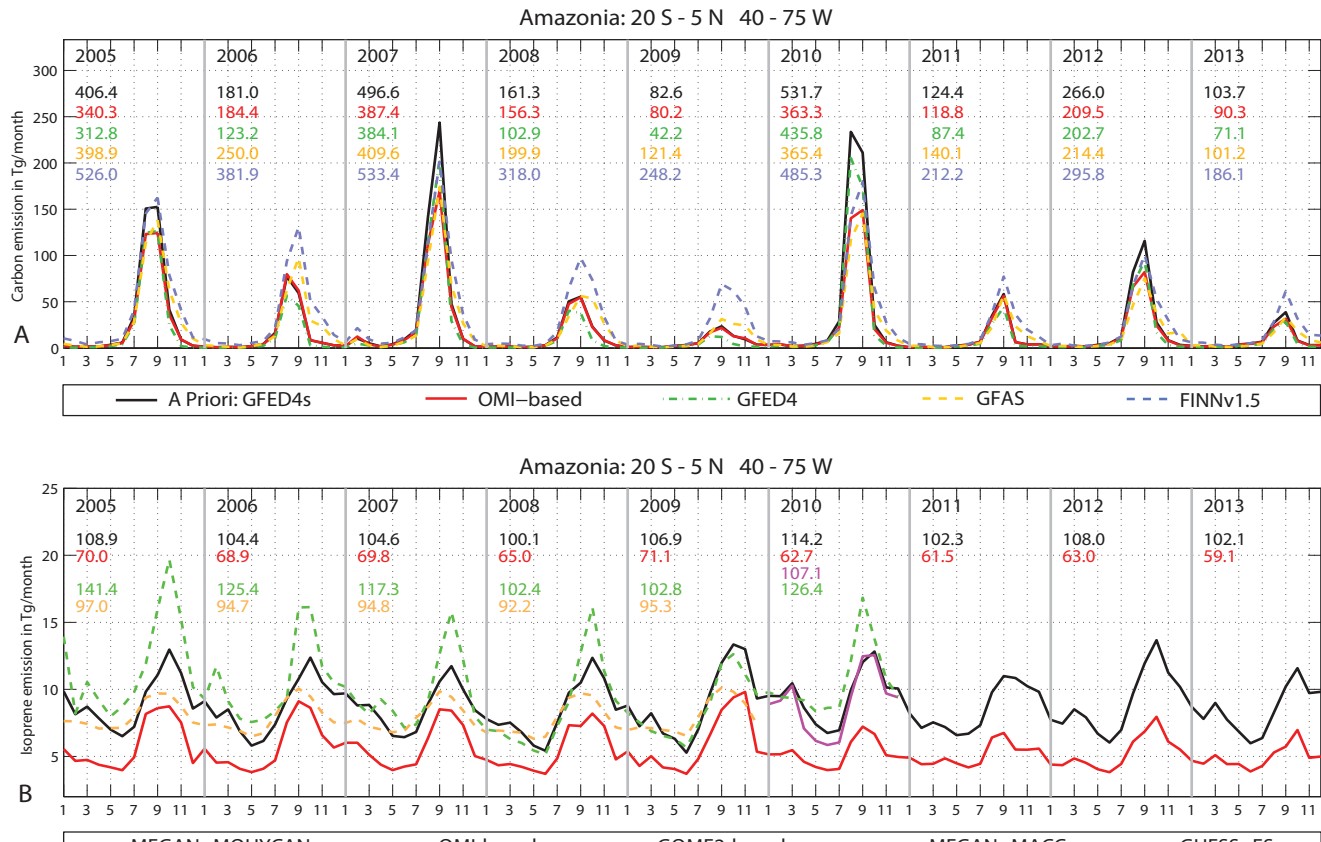

**Figure 9.** Seasonal and interannual variation of biomass burning emissions and isoprene emissions from bottom-up and top-down estimates over Amazonia (Fig. 3). Units are TgC per month for biomass burning fluxes and Tg of isoprene per month for biogenic emissions. The annual emission flux per inventory is given inset.





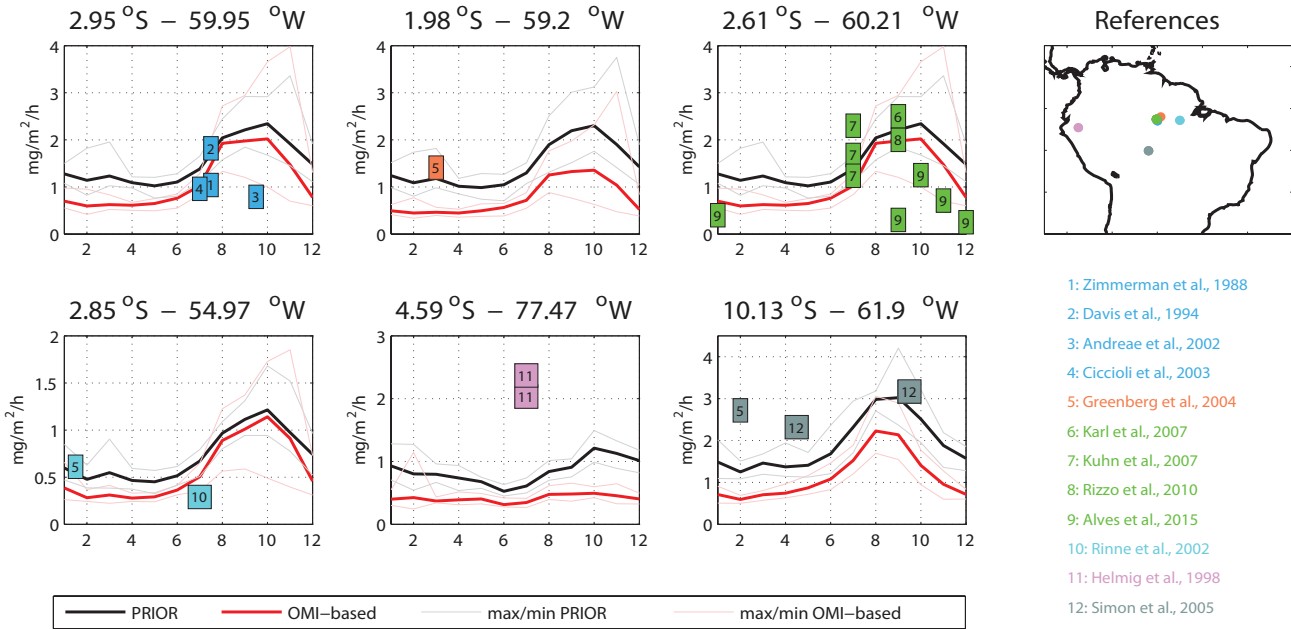

**Figure 10.** Comparison of a priori (black) and satellite-based (red) isoprene fluxes with ground-based flux measurements (colored numbered squares). The a priori and a posteriori isoprene fluxes are averaged over the full period from 2005 to 2013 for the grid. To ensure meaningful comparison, the ground-based flux measurements are corrected for the diurnal variation in isoprene fluxes, cf. Table S1 for more details.





**Figure 11.** As Fig. 9 for Africa.





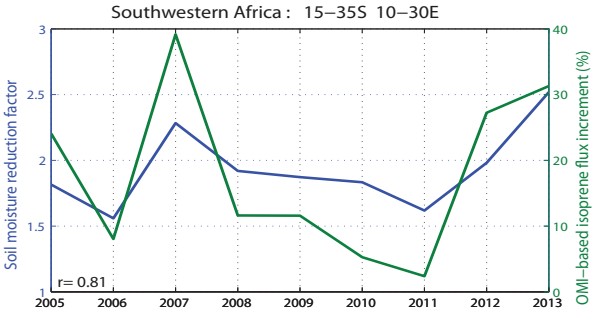

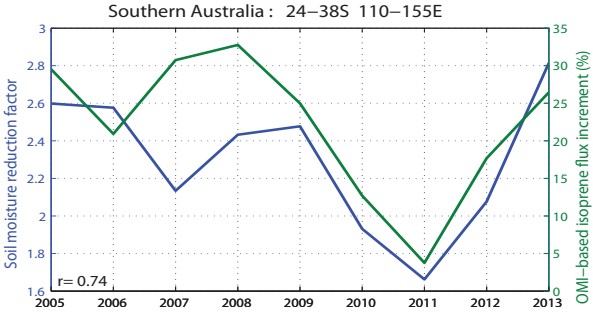

**Figure 12.** Interannual evolution of the factor by which the annual isoprene flux is reduced due to soil moisture stress vs. the isoprene flux increment inferred from OMI data (in %) in Southwest Africa (top) and Southern Australia (bottom).



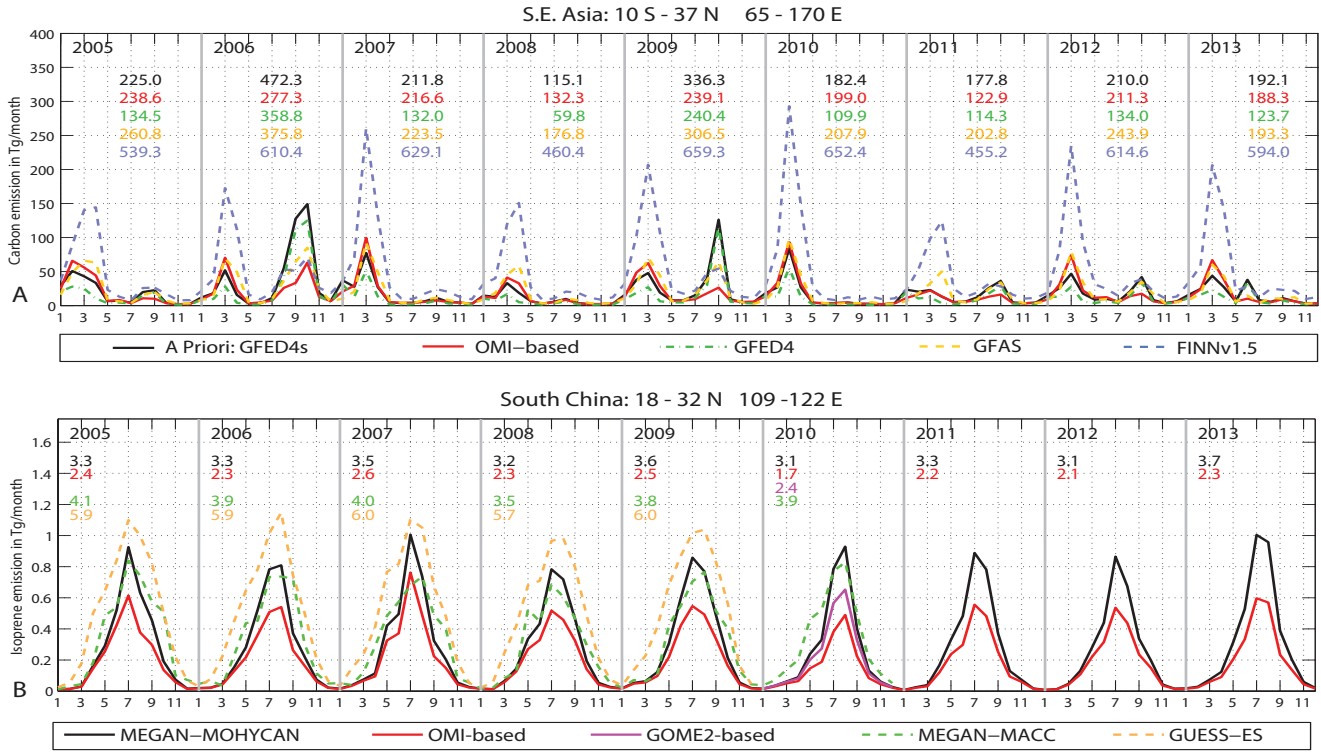

**Figure 13.** As Fig. 9 for Southeast Asia.





**Figure 14.** As Fig. 9 for Northern and Southern Australia.





**Figure 15.** As Fig. 9 for Europe and Southeast US.





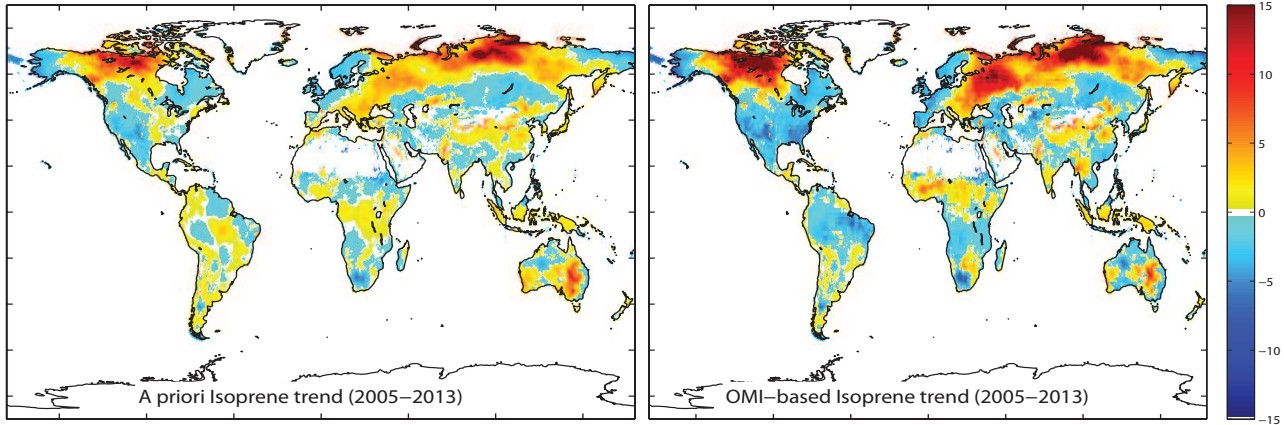

**Figure 16.** Global distribution of annual isoprene emission trends over 2005-2013 according to the a priori (left) and top-down inventory (right) expressed in %/yr.



**Table 1.** Mean a priori and OMI-based emission estimates compared to independent emission inventories for open biomass burning and isoprene emissions calculated for different world regions and globally. Regions are defined in Fig. 3. The means are taken over the period of data availability, i.e. over 2005-2013 for all inventories, except for MEGAN-MACC (2005-2010) and GUESS-ES (2005-2009).

| | North America | South America | Europe | NH Africa | SH Africa | Russia | Southeast Asia | Australia | **Global** |
|---|---|---|---|---|---|---|---|---|---|
| Biomass burning emissions (burnt biomass in TgC/yr) | | | | | | | | | |
| GFED4s | 105 | 319 | 31 | 418 | 684 | 130 | 237 | 104 | **2028** |
| OMI-based | 86 | 273 | 35 | 320 | 530 | 112 | 203 | 95 | **1653** |
| GFAS | 187 | 328 | 22 | 333 | 431 | 264 | 246 | 126 | **1938** |
| FINNv1.5 | 112 | 452 | 34 | 278 | 415 | 114 | 579 | 22 | **2006** |
| GFED4 | 84 | 231 | 17 | 279 | 479 | 97 | 156 | 95 | **1438** |
| Isoprene emissions (Tg isoprene/yr) | | | | | | | | | |
| MEGAN MOHYCAN | 32 | 141 | 6.8 | 50 | 29 | 9.4 | 36 | 38 | **343** |
| OMI-based | 26 | 97 | 8.4 | 35 | 28 | 11 | 31 | 36 | **272** |
| MEGAN-MACC | 34 | 173 | 7.8 | 103 | 67 | 12 | 80 | 94 | **570** |
| GUESS-ES | 44 | 143 | 18.1 | 77 | 60 | 20 | 63 | 26 | **452** |



**Table 2.** Global a priori and OMI-based emission estimates per year. Fire estimates are expressed in TgC/yr, isoprene in Tg of isoprene per year.

| Year | A priori fires | Optimized fires | A priori isoprene | Optimized isoprene |
|---|---|---|---|---|
| 2005 | 2252 | 1936 | 349 | 282 |
| 2006 | 2207 | 1721 | 339 | 280 |
| 2007 | 2202 | 1966 | 340 | 285 |
| 2008 | 1873 | 1605 | 324 | 263 |
| 2009 | 1862 | 1504 | 339 | 269 |
| 2010 | 2150 | 1679 | 363 | 272 |
| 2011 | 1872 | 1404 | 341 | 258 |
| 2012 | 2058 | 1676 | 350 | 276 |
| 2013 | 1773 | 1383 | 338 | 265 |
| 2005-2013 | 2028 | 1653 | 343 | 272 |



**Table 3.** Temporal Correlation between monthly MODIS fire counts, GFED4s and OMI-based fluxes over the regions selected based on literature evidence for the occurrence of small fires. The regions are shown on the MODIS land cover map in Fig. S4. Notes : [a] Region dominated by cropland according to the MODIS land cover change (Justice et al., 2002); [b] Region with a high number of small deforestation fires (Chen et al., 2013); [c] Region with peat fires selected based on Andela et al. (2013); [d] Region with a high number of small deforestation fires (Karki, 2002); [e] Region where GFED4s emissions are predominantly associated to small fires (Randerson et al., 2012).

| Region | Coordinates | Fire type | MODIS vs. GFED4s | MODIS vs. OMI-based |
|---|---|---|---|---|
| N. Africa | 4-16 N, 15W-15 E | agricultural [a] | 0.89 | 0.96 |
| Maranhão | 6S-2 N, 44-52 W | agricultural[a] | 0.56 | 0.91 |
| Mato Grosso | 7-15 S, 50-60 W | small scale deforestation[b] | 0.95 | 0.97 |
| SE US | 30-36 N, 75-100 W | agricultural[a] | 0.36 | 0.65 |
| N. China | 30-40 N,111-122 E | agricultural[a] | 0.66 | 0.85 |
| Indochina | 6-27 N, 87-110 E | agricultural[a] and small-scale deforestation[d] | 0.84 | 0.95 |
| Indonesia | 10 S-5 N,93-130 E | agricultural[a] and peat[c] | 0.85 | 0.89 |
| NW India | 29-33 N,70-79 E | agricultural fires[a] | 0.75 | 0.87 |
| Russia | 52-60 N, 55-90 W | agricultural[a] and peat[c] | 0.81 | 0.94 |
| Eq. Africa | 14 S-2 N, 10-25 E | agricultural[e] | 0.96 | 0.99 |
| E. Australia | 20-40 S, 145-155 E | agricultural[e] | 0.55 | 0.86 |
| Madagascar | 12-26 S, 43-50 E | agricultural[e] | 0.90 | 0.96 |