# Peer review of "Nine years of global hydrocarbon emissions based on source inversion of OMI formaldehyde observations"

_Atmospheric Chemistry and Physics, 2016_

## Referee Comment (RC1) · Anonymous Referee #1 · 27 Apr 2016

In my opinion, this is a great paper, and surely a benchmark in the field. Although it is long!

It presents a thorough discussion of the inferred top-down fire and isoprene emissions, and compares them both to independent emissions inventories and flux measurements (for isoprene). Figures are excellent (though units should really be attached to colour bars in figs 1, 2, 5, 6, 16)

If I had one issue, and this is maybe really for a future paper, it would be to compare the simulated IMAGES tracers based on the a-priori and optimised emissions against observations. That it is actually compare the simulated concentrations of isoprene, HCHO + other key VOCs and tracers against in-situ ground and aircraft observations

to really see how the model improves. Comparing with other emission estimates is good, but you really want to see if the model does better in simulating atmospheric chemistry. There are plenty of observational datasets during the studied time period to do this.

Minor comments.

Abstract: first sentence does sound right when you read it. Maybe: 'As HCHO is a high yield...'

OMI row anomaly - did you check how the number of observations changes per grid cell, and how that correlates with inferred emission trends. Did you also try only the OMI rows 5-23 which are unaffected throughout the mission?

page 5, line 24: 'Inversions are performed separately for each year' - in the framework of a continuous adjoint simulation? i.e was it a start-stop inversion?

page 6, line 23: there is a '(s)' -is this a typo?

———————————————————

---

## Referee Comment (RC2) · Anonymous Referee #2 · 29 Apr 2016

Bauwens et al. present an analysis of nine years of global hydrocarbon emissions inferred from OMI formaldehyde observations. 2005-2013 global distributions of pyrogenic and biogenic VOC fluxes are derived from OMI HCHO columns and the adjoint inversion scheme based on the IMAGESv2 global CTM. The distributions, their interannual and seasonal variations are discussed for the different regions where the changes are the most important compared to the a priori emissions. The inversed emission fluxes are compared and discussed according to the various independent inventories. Trends over the studied period are derived and discussed.

The paper is well written and structured with detailed discussions of the major changes compared to the a priori in terms of distribution, seasonality and interannual variability. Trends are also well documented and discussed. This work is suitable for ACP publication and I recommend it after the following specific comments are addressed.

Specific comments:

- Page 5, lines 25-26: it should be interesting to give a range of retrieval errors to see how it compares to the representativity error.

- Page 5, lines 26-27: On which studies, references the a priori error estimate of biogenic and pyrogenic fluxes is based? Please, discuss this point.

- Page 6, lines 1-3: Some of the regions are not covered by the observations depending the season. How does this impact on the retrieved fluxes? Is the induced uncertainty can be estimated?

- Pages 7-8 – discussion of Figure 4: Europe presents a large interannual variability, which is not discussed in this section. Please, add some explanations here or mention it and refer to the corresponding section (section 8) if suitable.

- Page 9, lines 27-28: it is not clear how agricultural fires can be maximum in December while they are dominant for other periods of the year. Could you make it clearer, please?

- Figure 1: it would help to provide difference plots

Technical comments:

- Page 5, lines 13-15: the sentence is too long and should be rephrased for clarity.

- Page 10, line 28: remove "in Northern Africa" after "the isoprene fluxes".

- Page 16, line 27: there are two "and" close to the end of the line. Remove one.

---

## Author Comment (AC1) · 23 Jun 2016

**Reply to anonymous Referee#1**

We would like to thank the reviewer for his/her positive evaluation of the manuscript and for the useful comments and suggestions. Below we address the raised concerns. The reviewer's comments are *italicized*.

*In my opinion, this is a great paper, and surely a benchmark in the field. Although it is long! It presents a thorough discussion of the inferred top-down fire and isoprene emissions, and compares them both to independent emissions inventories and flux measurements (for isoprene).*

*Figures are excellent (though units should really be attached to color bars in figs 1, 2, 5, 6, 16)*

Units are added in Fig. 1, 2, 5, 6, 16, as well as in supplementary figures S2, S3 and S4.

*If I had one issue, and this is maybe really for a future paper, it would be to compare the simulated IMAGES tracers based on the a-priori and optimized emissions against observations. That it is actually compare the simulated concentrations of isoprene, HCHO + other key VOCs and tracers against in-situ ground and aircraft observations to really see how the model improves. Comparing with other emission estimates is good, but you really want to see if the model does better in simulating atmospheric chemistry. There are plenty of observational datasets during the studied time period to do this.*

As the paper is already long, we limited the validation to comparison with available isoprene flux measurements over the Amazon (Section 4). The modelled formaldehyde concentrations based on a priori and optimized emissions were evaluated against aircraft observations over North America in our previous inverse modelling study based on OMI (and GOME-2) HCHO columns (Stavrakou et al., 2015). Nevertheless, we agree with the referee that in a future study the satellite-based emissions and model concentrations for other compounds should be validated against available observations.

*Minor comments*

*Abstract: first sentence does sound right when you read it. Maybe: "As HCHO is a high yield…"*

Corrected.

*OMI row anomaly - did you check how the number of observations changes per grid cell, and how that correlates with inferred emission trends.*

As explained in our answer to the next comment, we omitted from the calculation of the monthly OMI column averages the rows that disappear in 2013 throughout the entire OMI time series (2005-2013), in order to minimize the effect of the OMI row anomaly on the estimated trends. Nevertheless, it is a good suggestion by the referee to check whether the changes in the number of observations correlate with the inferred emission trends. The following figure shows that the

annually averaged HCHO columns (VCD, in red) are indeed positively correlated with the number of observations (NBP, in blue). Furthermore, despite our removal of anomalous OMI rows, the number of observations shows a decreasing trend in many regions, likely reflecting a slow deterioration of the OMI retrievals unrelated to the row anomaly. As already pointed out in the manuscript, it cannot be ruled out that time-dependent instrumental effects might impact the inferred emission trends.

The strong correlation between the number of observations and the OMI columns displayed in the figure would be very worrisome if it could not be easily explained by the important role of clouds in both the HCHO retrievals and the formation of HCHO in the atmosphere. Because of our cloud filter excluding scenes with more than 40% cloud fraction, cloudy conditions are associated to fewer observations, as shown by the good correlation of NBP with the PAR (Photosynthetically Active Radiation) irradiance at the surface, obtained from the ECMWF ERA-Interim analysis. The HCHO columns are also usually depressed under cloudy conditions, primarily due to the influence of light on biogenic VOC emissions. PAR is also correlated with temperature (which also drives BVOC emissions) and with the abundance of the OH radicals which is the primary VOC oxidizing agent. As a result, PAR is found to be well-correlated with the HCHO columns over most regions, e.g. over the Southeastern U.S. (0.73) and over Amazonia (0.83). Although there appears to be a clear declining PAR trend over Western Europe between 2005 and 2013, there is no discernible PAR trend over Amazonia and the S-E U.S. A trend in cloudiness is therefore not responsible for the negative emission trends over these regions.

[Figure]

Figure. Normalized mean annual number of observations (NBP, blue) and OMI HCHO columns (VCD, red) over Southeastern US (26-36 N, 75-100 W), Western Europe (37-71 N, 10W-20 E), Amazonia (5-20 S, 40-75 W) and South China (18-32 N, 109-122 E) between 2005 and 2013. Annual mean photosynthetically active radiation (PAR) in the same regions is shown in green. The correlations between PAR and NBP ($r_{NBP}$) and between PAR and VCD ($r_{VCD}$) are shown inset.

Did you also try only the OMI rows which are unaffected throughout the mission?

As mentioned in the manuscript, the filtering of the row anomaly leads to a loss of coverage throughout the years. To ensure meaningful determination of trends, we omitted from the calculation of the monthly OMI HCHO column averages the rows that disappear in 2013 (rows 28-46, 54 and 55) throughout the entire OMI time series (2005-2013). The choice results in a reduction in spatial coverage of the OMI data since the beginning of the mission, but ensures a constant spatial coverage in 2005-2013 and pixels of the same size. The effect of this filtering is a slight decrease of the column means in the beginning of the time series.

*page 5, line 24: "Inversions are performed separately for each year" - in the framework of a continuous adjoint simulation? i.e was it a start-stop inversion?*

We performed 9 independent inversions for each year of the study period, each constrained by satellite observations of the corresponding year. The inversions start in January after a 4-month spin-up. In each inversion, the model is confronted with 12 months of OMI observations, after a 6 month spin-up.

*page 6, line 23: there is a '(s)' -is this a typo?*

The (s) denotes the GFED4s inventory which includes small fires. To avoid confusion we removed it.

---

## Author Comment (AC2) · 23 Jun 2016

**Reply to anonymous Referee#2**

We would like to thank the reviewer for his/her positive evaluation of the manuscript and for the useful comments and suggestions. Below we address the raised concerns. The reviewer's comments are *italicized* and our replies are given in blue.

*Bauwens et al. present an analysis of nine years of global hydrocarbon emissions inferred from OMI formaldehyde observations. 2005-2013 global distributions of pyro genic and biogenic VOC fluxes are derived from OMI HCHO columns and the adjoint inversion scheme based on the IMAGESv2 global CTM. The distributions, their interannual and seasonal variations are discussed for the different regions where the changes are the most important compared to the a priori emissions. The inversed emission fluxes are compared and discussed according to the various independent inventories. Trends over the studied period are derived and discussed. The paper is well written and structured with detailed discussions of the major changes compared to the a priori in terms of distribution, seasonality and interannual variability. Trends are also well documented and discussed. This work is suitable for ACP publication and I recommend it after the following specific comments are addressed.*

*Specific comments:*

*- Page 5, lines 25-26: it should be interesting to give a range of retrieval errors to see how it compares to the representativity error.*

The retrieval error amounts to about 40-60% of the OMI column over emission regions and ranges between $4-7 \times 10^{15}$ molec.cm$^{-2}$. The assumed representativity error of $2 \times 10^{15}$ molec.cm$^{-2}$ is generally low in comparison, and has little effect on the total error because of the geometric summation.

*- Page 5, lines 26-27: On which studies, references the a priori error estimate of biogenic and pyrogenic fluxes is based? Please, discuss this point.*

We acknowledge that the precise values of the flux error estimates are very uncertain. The factor of 3 error on the a priori biogenic and pyrogenic fluxes reflects the high variability of the pyrogenic emission source and the strong uncertainties associated with the biogenic emissions, as demonstrated by the large range of literature emission estimates (Sinderalova et al., 2014, Arneth et al. 2011). This is now added in the revised manuscript. The sensitivity of our inversions to the chosen error estimates was evaluated by means of sensitivity inversions in Stavrakou et al. (2015).

*- Page 6, lines 1-3: Some of the regions are not covered by the observations depending the season. How does this impact on the retrieved fluxes? Can the induced uncertainty be estimated?*

When top-down constraints are missing over a region or month, the a posteriori emissions remain generally close to the a priori, although they are affected by neighboring emission regions,

because of transport of formaldehyde precursors, and because of the spatiotemporal correlations implemented in the inversion scheme.

*- Pages 7-8 – discussion of Figure 4: Europe presents a large interannual variability, which is not discussed in this section. Please, add some explanations here or mention it and refer to the corresponding section (section 8) if suitable.*

We now mention that the interannual variability of the optimized emissions is thoroughly discussed in Sec. 4, 5, 6, and 8.

*- Page 9, lines 27-28: it is not clear how agricultural fires can be maximum in December while they are dominant for other periods of the year. Could you make it clearer, please?*

In this region, both natural and agricultural fires peak in December, but the agricultural fire season, from September to May, lasts longer than the season of natural fires, which occurs between November and March. Therefore, in the beginning and end of the fire season the agricultural fires are dominant (Magi et al., 2012). This is now clarified in the manuscript.

*- Figure 1: it would help to provide difference plots*

The figure has been adapted.

*Technical comments:*

*- Page 5, lines 13-15: the sentence is too long and should be rephrased for clarity.*

Clarified.

*- Page 10, line 28: remove "in Northern Africa" after "the isoprene fluxes".*

Removed.

*- Page 16, line 27: there are two "and" close to the end of the line. Remove one.*

Corrected.

---

## Author Response (AR1)

Royal Belgian Institute for Space Aeronomy Department "Atmospheres" Section "Sources and Sinks" Avenue Circulaire 3, B-1180 Brussels, Belgium

Brussels, 23 June 2016

Dear Editor,

The comments and technical corrections suggested by the reviewers are addressed in the submitted replies to the reviewers. The changes in the manuscript are very small. We would like to thank you again for your consideration.

Yours sincerely,

Dr. Trissevgeni Stavrakou Trissevgeni.Stavrakou@aeronomie.be

frayedat

**Nine years of global hydrocarbon emissions based on source inversion of OMI formaldehyde observations**

Maite Bauwens1, Trissevgeni Stavrakou1, Jean-François Müller1, Isabelle De Smedt1, Michel Van Roozendael1, Guido R. van der Werf2, Christine Wiedinmyer3, Johannes W. Kaiser4, Katerina Sindelarova5,6, and Alex Guenther7

1Royal Belgian Institute for Space Aeronomy, Avenue Circulaire 3, 1180, Brussels, Belgium
2Vrije Universiteit Amsterdam, Faculty of Earth and Life Sciences, Amsterdam, The Netherlands
3National Centre for Atmospheric Research, Boulder, CO, USA
4Max Planck Institute for Chemistry, Mainz, Germany

[revised manuscript text omitted]